# THE MIND'S TRANSFORMER: COMPUTATIONAL NEUROANATOMY OF LLM-BRAIN ALIGNMENT

**Cheng-Yeh Chen, Raghupathy Sivakumar**
Georgia Institute of Technology, Atlanta, GA, USA
cchen847@gatech.edu, siva@ece.gatech.edu

## ABSTRACT

The alignment of Large Language Models (LLMs) and brain activity provides a powerful framework to advance our understanding of cognitive neuroscience and artificial intelligence. In this work, we zoom into one of the fundamental units of LLMs—the transformer block—to provide the first systematic computational neuroanatomy of its internal operations and human brain acitivity during language processing. Analyzing 21 state-of-the-art LLMs across five model families, we extract and evaluate 13 distinct intermediate states per transformer block—from initial layer normalization through attention mechanisms to feed-forward networks (FFNs). Our analysis reveals three key findings: (1) The commonly used hidden states in LLMs are surprisingly suboptimal, with over 90% of brain voxels in sensory and language regions better explained by previously unexplored intermediate computations; (2) Different computational stages within a single transformer block map to anatomically distinct brain systems, revealing an intra-block hierarchy where early attention states align with sensory cortices while later FFN states correspond to association areas—mirroring the cortical processing hierarchy; (3) Rotary Positional Embeddings (RoPE) specifically enhance alignment along the brain's auditory processing streams. Per-head queries with RoPE best explain 74% of auditory cortex activity compared to 8% without RoPE, providing the first neurobiological validation of this architectural component in LLMs. Building on these insights, we propose MindTransformer[1], a feature selection framework that learns brain-aligned representations from all intermediate states. MindTransformer achieves significant brain alignment performance, with correlation improvements in primary auditory cortex exceeding gains from 456× model scaling. Our computational neuroanatomy approach opens new directions for understanding both biological intelligence through the lens of transformer computations and artificial intelligence through principles of brain organization.

## 1 INTRODUCTION

The remarkable success of Large Language Models (LLMs) has catalyzed a fundamental question in cognitive science: do these artificial systems process language through mechanisms similar to the human brain? Recent work has demonstrated striking correlations between LLM representations and neural activity measured through fMRI (Toneva & Wehbe, 2019; Schrimpf et al., 2021; Caucheteux & King, 2022; Goldstein et al., 2022), with alignment improved through various interventions including dataset scaling (Antonello et al., 2023; Gokce & Schrimpf, 2025; Ren et al., 2025), model scaling (Antonello et al., 2023; Gokce & Schrimpf, 2025; Ren et al., 2025; Bonnasse-Gahot & Pallier, 2024), prompting (Sun & Moens, 2023; Ren et al., 2025), fine-tuning (Sun & Moens, 2023; Aw et al., 2024; Oota et al., 2025), and taskonomy (Oota et al., 2022; Aw & Toneva, 2023). These findings suggest that LLMs may serve as computational models of human language processing, with applications ranging from brain prediction (d'Ascoli et al., 2026) to causal manipulation of neural activity (Tuckute et al., 2024).

However, a critical methodological limitation undermines current understanding: existing approaches treat LLM architectures as black boxes, using only a single representation per layer while

---

[1]Source code: https://github.com/cheng-yeh/MindTransformer

overlooking the rich internal computations within LLMs. This practice assumes that all neurally-relevant information is compressed into a single vector, ignoring the intermediate computations—the layer normalization, positional embeddings, multi-head attention projections, and feed-forward transformations—that collectively implement the model's processing. While pioneering studies have examined specific components like attention weights (Lamarre et al., 2022) or individual attention heads (Kumar et al., 2024), no systematic analysis exists of how all intermediate computations inside transformer blocks map to brain activity. This gap obscures potential convergence points between the discrete, semantic nature of text processing and the continuous, sensory-driven mechanisms of the brain. While acoustic language models have successfully characterized neural activity in low-level auditory regions (Tuckute et al., 2024; Millet et al., 2022; Antonello et al., 2023), text-based LLMs typically fail to achieve meaningful alignment in these areas, suggesting a disconnect between textual and sensory representations (Caucheteux & King, 2022; Kauf et al., 2024; Doerig et al., 2025; Goldstein et al., 2025). Consequently, the neurobiological relevance of key architectural components that bridge this gap—such as positional embeddings—remains unexplored, leaving a disconnect between engineering design choices and biological plausibility.

We address these challenges through a comprehensive computational neuroanatomy of transformer block architectures, one of the fundamental units for LLMs. Our approach systematically decomposes each transformer block into 13 distinct intermediate states—from pre-attention normalization through per-head attention computations to feed-forward network activations—and evaluates their individual and collective correspondence with brain activity. Analyzing 21 state-of-the-art models (270M to 123B parameters) from five major families (Llama, Qwen, Mistral, GPT, Gemma) on naturalistic story listening fMRI data, we make three principal contributions:

- **Revealing the suboptimality of current practices through systematic analysis.** We demonstrate that the commonly used hidden states are remarkably inefficient, with over 90% and 96% of brain voxels in language and sensory regions better explained by previously unexplored intermediate computations. This finding provides a new dimension to align neurally-relevant representation in LLMs onto the brain.

- **Uncovering an intra-block processing hierarchy that mirrors cortical organization.** Different computational stages within a single transformer block map to anatomically distinct brain systems—early attention states align with sensory cortices while later FFN states correspond to association areas. This reveals a fine-grained computational hierarchy within each block that parallels the brain's own anatomical processing hierarchy, extending beyond the known layer-wise progression in LLMs.

- **Establishing robust alignment with low-level sensory processing through architectural components.** We identify that Rotary Positional Embeddings (RoPE) specifically enhance alignment with the brain's auditory processing streams. Inside the multi-head attention, per-head queries with RoPE best explain 74% of auditory cortex voxels versus 8% without, systematically improving alignment along both ventral and dorsal auditory pathways. This provides the first neurobiological validation of RoPE's functional role and demonstrates that architectural design choices can have direct neural correlates.

Building on these insights, we propose MindTransformer, a principled framework that learns brain-aligned representations from all intermediate states. First, it discovers neurally-relevant features through ridge regression on concatenated representations; second, it selects the most informative subset for final model training. MindTransformer achieves significant performance in language network and audio cortex, especially with correlation improvements of 0.111 in primary auditory cortex—gains that exceed those from scaling LLMs by 456× (from 270M to 123B parameters).

## 2 RELATED WORK

### 2.1 THE LANDSCAPE OF LLM-BRAIN ALIGNMENT

Research in LLM-brain alignment has established that model representations increasingly correspond with neural activity as models and datasets scale (Antonello et al., 2023; Gokce & Schrimpf, 2025; Ren et al., 2025). This scaling allows models to recover fundamental brain properties like left-hemisphere lateralization (Bonnasse-Gahot & Pallier, 2024). Alignment is further improved by

training models on specific objectives, such as cognitively demanding tasks (Oota et al., 2022; Aw & Toneva, 2023) or instruction-following (Aw et al., 2024; Oota et al., 2025). Recent work has also explored prompt engineering strategies to enhance alignment (Sun & Moens, 2023; Ren et al., 2025). The strength of this connection has enabled applications ranging from brain prediction (d'Ascoli et al., 2026) to causal control of neural activity (Tuckute et al., 2024). Due to much better alignment in regions of high-level semantic processing in the brain, these preceding endeavors form the prevailing view that LLM-brain alignment converges to high-level semantic processing, with low-level sensory regions remaining inaccessible (Caucheteux & King, 2022; Kauf et al., 2024; Doerig et al., 2025; Goldstein et al., 2025).

## 2.2 FROM MONOLITHIC TO MECHANISTIC ALIGNMENT

While most alignment research uses the final hidden state, a growing body of work has begun to probe specific transformer mechanisms, with a strong focus on the attention component. This line of inquiry has shown that raw attention weights (Lamarre et al., 2022), their similarity to human eye-tracking patterns (Gao et al., 2023), and the specialized computations of individual attention heads (Kumar et al., 2024) are all predictive of distinct neural activity. Intriguingly, even shallow, untrained attention networks exhibit brain-like properties, highlighting the importance of architectural biases (AlKhamissi et al., 2024).

These pioneering studies validate examining internal mechanisms but leave critical gaps. First, they focus narrowly on attention while ignoring other computational stages—the layer normalization, positional embeddings, and feed-forward networks that comprise the majority of transformer computations. Second, they lack systematic analysis across the full range of intermediate states within transformer blocks. Our work provides the first comprehensive computational neuroanatomy analysis across all intermediate computations, revealing how each component contributes to brain alignment and demonstrating that the choice of representation fundamentally determines which brain systems can be modeled.

## 3 METHODOLOGY

### 3.1 BRAIN DATASET AND PREPROCESSING

We use the publicly available *Le Petit Prince fMRI Corpus* (Li et al., 2022), a dataset specifically designed for studying the neural basis of language during naturalistic story listening. The corpus provides fMRI data for three languages and we use the English subset with 49 English native speakers. During fMRI acquisition, each participant listened to an audiobook of the story *Le Petit Prince*, which is approximately 100 minutes in duration and structurally divided into 9 distinct runs.

To create a robust, group-level signal and improve the signal-to-noise ratio (SNR), we average the fMRI time-series across all participants[2]. This group-averaged signal was used for all subsequent encoding models, with regions of interest (ROIs) defined using the Harvard-Oxford structural atlas from Jenkinson et al. (2012) or the language localizer from Fedorenko et al. (2010) to enable precise computational neuroanatomy mapping.

### 3.2 LLM ACTIVATION EXTRACTION AND PREPROCESSING

To comprehensively investigate the computational neuroanatomy of transformer architectures, we analyze 21 state-of-the-art open-weight LLMs spanning five major model families: Llama, Qwen, Mistral, GPT, and Gemma. These models range from 270M to 123B parameters, encompassing diverse architectural choices including standard multi-head attention (MHA), grouped-query attention (GQA), multi-query attention (MQA), and mixture-of-experts (MoE) architectures (see Appendix A for complete architectural specifications).

We process the corresponding stimulus text in the fMRI dataset through all 21 models. The text is first tokenized using each model's native tokenizer, then passed through the model to extract intermediate activations. We extract activations from every transformer block across all layers. For

---

[2]All the preprocessing on fMRI, including downsampling, averaging, masking, etc. is available in our source code (https://github.com/cheng-yeh/MindTransformer).

a model with $N_{\text{layer}}$ transformer blocks, this yields $N_{\text{layer}} \times 13$ distinct activation states per token, corresponding to the 13 intermediate states we identify within each block (detailed in Section 3.4).

To align these token-level activations with the temporal resolution of fMRI, we implement a word-level aggregation strategy commonly used in literature (Antonello et al., 2023; Bonnasse-Gahot & Pallier, 2024). Since transformer tokenizers often split words into subword tokens, we average the activation vectors of all tokens belonging to the same word. This produces a sequence of word-level activation vectors directly corresponding to the word-by-word onset timing in the fMRI experiment.

## 3.3 Voxel-wise Encoding Pipeline

Our work builds upon the standard voxel-wise linear encoding paradigm, a widely adopted and validated methodology for aligning LLM representations with fMRI data (Antonello et al., 2023; Kumar et al., 2024; Tuckute et al., 2024). This pipeline learns a mapping from LLM features to fMRI signals for each brain voxel, forming the foundation of our computational neuroanatomy approach. Our implementation proceeds as follows[3]:

1. **HRF Convolution:** The brain's hemodynamic response measured by fMRI is slow and delayed relative to neural activity. To account for this neurobiological constraint, we convolve the time-series of word-level LLM activations (obtained from the preprocessing step above) with a canonical Hemodynamic Response Function (HRF)—specifically the Glover HRF (Glover, 1999)—to create a feature space that is temporally aligned with fMRI signal.

2. **Ridge Regression:** We use L2-regularized linear regression (Ridge Regression) to learn a mapping from the HRF-convolved LLM features to the fMRI signal for each individual brain voxel. The regularization parameter $\alpha$ is optimized via nested cross-validation within the training folds, typically ranging from $10^{-2}$ to $10^4$.

3. **Robust Cross-Validation:** To ensure generalizability of our computational neuroanatomy findings, we employ a 9-fold cross-validation scheme based on the dataset's 9 runs. For each fold, one run is held out as the test set, and the model is trained on the remaining 8 runs. This guarantees that the test set is always entirely unseen during training[4]. The final reported performance is the Pearson correlation between the predicted and actual fMRI signals, averaged across the 9 folds of testing.

## 3.4 Dissecting the Transformer Block

Rather than using a single representation, we decompose the transformer's internal computation into thirteen distinct states that capture the complete information processing pipeline. This granular approach reveals how different stages of computation correspond to distinct neuroanatomical systems.

We organize these states into three major computational stages that reflect universal computations of transformer blocks for most state-of-the-art LLMs:

1. **Block Input** consists of the input hidden state from the previous layer and its pre-attention normalized state that stabilizes activations for subsequent processing.

2. **Attention Mechanism** encompasses seven critical states: per-head queries and keys both before and after applying Rotary Positional Embeddings (RoPE)—allowing us to isolate the neural contribution of positional encoding; per-head values containing the content to be attended to; per-head context vectors representing each head's weighted synthesis; and the combined attention output after projection.

3. **FFN & Residuals** captures the remaining transformation pipeline: the post-attention hidden state after the first residual connection, its pre-FFN normalized state, the FFN activated state after expansion to higher dimension, and the FFN output after down-projection back to model dimension.

---

[3]All scripts for HRF covolution, ridge regression, and cross-validation are available in our source code (https://github.com/cheng-yeh/MindTransformer).

[4]The analysis in Section 4 selects optimal states based on test set performance for exploratory purposes. Conversely, the results in Section 5 enforce strict separation, with feature selection performed solely on the training set to prevent data leakage.

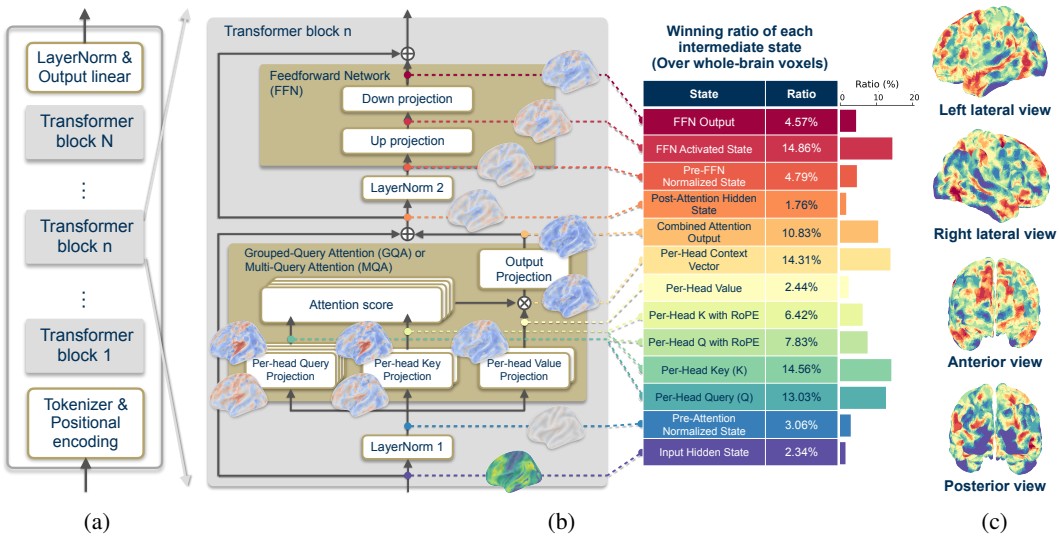

Figure 1: 13 intermediate states extracted from each transformer block inside LLMs. (a) Overall architecture of LLMs. (b) Per-state visualization of position within a transformer block and statistics of winning ratio from each intermediate state, averaged from all the transformer blocks of 21 LLMs in 5 families. (c) Brain plots colored by the best state for each voxel with various views for the 12-th layer of Llama-3.2 8B. The color map is provided in (b).

This systematic extraction yields a rich, multi-dimensional representation of the transformer's internal computation shown in Figure 1b. Each state is preserved with its full dimensionality, with tensor shapes ranging from $(B, S, D_{\text{model}})$ for pre-attention hidden states to $(B, N_q, S, D_{\text{head}})$ or $(B, N_{kv}, S, D_{\text{head}})$ for per-head attention-related representations and $(B, S, D_{\text{ffn}})$ for the expanded FFN activation, where $B$ is the batch size, $S$ is the sequence length, $D_{\text{model}}$ is the model dimension, $N_q$ and $N_{kv}$ are the number of attention heads for query and key/value, $D_{\text{head}}$ is the per-head dimension, and $D_{\text{ffn}}$ is the feed-forward inner dimension. We detail the definition of all intermediate states and their tensor dimensions in Appendix B.

## 4 COMPUTATIONAL NEUROANATOMY ANALYSIS

### 4.1 BEYOND THE HIDDEN STATE: AN EMERGENT INTRA-BLOCK HIERARCHY

Our first major finding in computational neuroanatomy fundamentally challenges the standard practice in LLM-brain alignment. We apply the voxel-wise encoding pipeline on each intermediate state to create a comprehensive computational neuroanatomy map. For each voxel, we identify the best intermediate state that could explain the activity of that voxel. As shown in Figure 1, the two most commonly used representations—the input hidden state and the per-head context vector (Kumar et al., 2024)—are suboptimal from a computational neuroanatomy perspective. Together, they best explain the activity in 16.65% (2.34% from input hidden state and 14.31% from per-head context vector in Figure 1b) of brain voxels. If we further focus on the audio-sensory cortex (Da Costa et al., 2011; Hamilton et al., 2021), or the language networks (Fedorenko et al., 2010) in the brain (regions of interest detailed in Appendix C), we get a lower percentage of best explained voxels. Merely 9.91% and 3.68% of voxels in the language network and audio cortex are best explained by the commonly used states (shown in Appendix D). The vast majority of the brain is better modeled by previously unexplored intermediate states, demonstrating the necessity of our granular computational neuroanatomy approach. Quantitatively, selecting the best intermediate state raises alignment correlations from 0.275 to 0.296 in the whole brain and from 0.433 to 0.450 in the language network, with the auditory cortex showing the largest improvement from 0.407 to 0.475 (Appendix Table 5).

Furthermore, our computational neuroanatomy analysis reveals a consistent pattern of functional specialization *within* the transformer block that mirrors the brain's own information processing hierarchy. Figure 2 shows that early-stage computations within a block, like attention-related states,

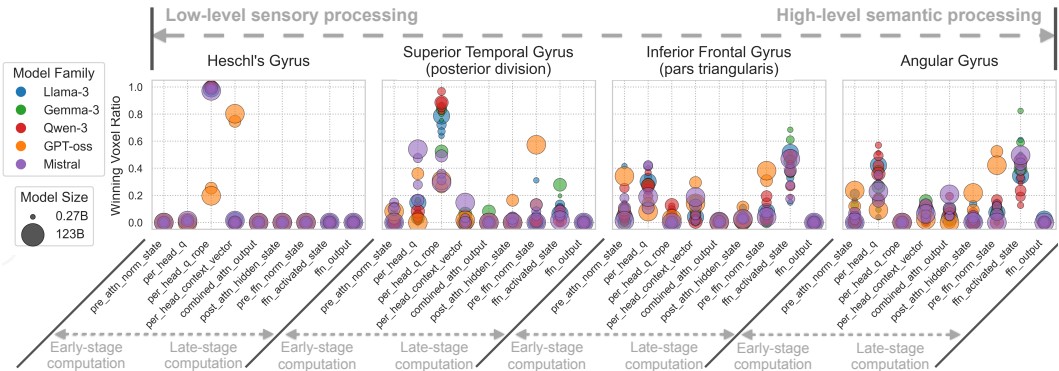

Figure 2: Functional specialization within a transformer block. Across LLM layers, early-stage states (e.g., per-head Query with RoPE) consistently dominate in low-level sensory regions (e.g., Heschl's gyrus and superior temporal gyrus), while late-stage states (e.g., FFN-related states) dominate in high-level association cortex (e.g., inferior frontal gyrus and angular gyrus), revealing an emergent intra-block processing hierarchy.

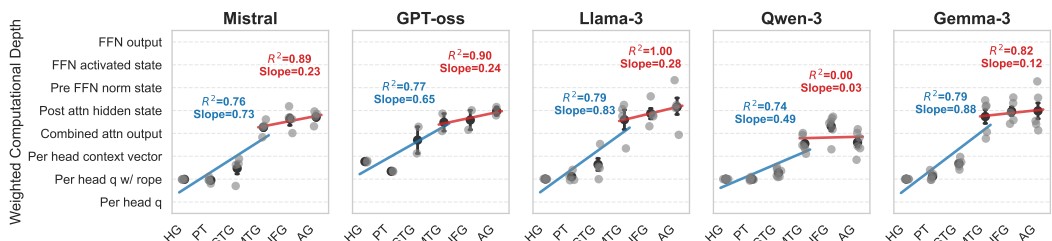

Figure 3: Weighted Computational Depth versus Cortical Hierarchy across LLM families. The analysis quantifies the topological alignment between the transformer's internal processing depth (y-axis) and the brain's cortical hierarchy (x-axis). We distinguish two functional segments: the auditory stream (HG to MTG) and the language network (MTG to AG). High $R^2$ values are observed across most LLM families, characterized by a steep slope in the auditory segment, indicating a rapid progression through early computational states, and a plateau in the language segment, confirming hierarchical mapping between biological auditory pathways and transformer block computations.

preferentially align with low-level sensory cortices (e.g., Heschl's gyrus and superior temporal gyrus), while late-stage computations, like FFN-related states, align with high-level association cortices (e.g., inferior frontal gyrus and angular gyrus). To quantify this, we define a metric of *Computational Depth*, calculated as the normalized index of the winning intermediate state within the transformer block's processing sequence. We correlate this with the *Cortical Hierarchy* of the corresponding brain regions. Detailed mathematical formulation of these metrics is provided in Appendix E. As shown in Figure 3, we observe a striking consistency across all LLM families. The auditory processing stream (spanning from Heschl's Gyrus to the Middle Temporal Gyrus) exhibits a steep positive slope, demonstrating a strong linear mapping where ascending cortical levels correspond to deeper intra-block computational depth. In contrast, the high-level language network (extending from the MTG to the Angular Gyrus) displays a flattened trajectory, indicating a computational plateau where alignment stabilizes at the block's later stages. This discovery suggests an intra-block computational hierarchy that parallels neuroanatomical organization: early attention-related states process immediate, stimulus-driven information akin to sensory cortices, while later FFN states handle more abstract, integrated information similar to association areas. This finding extends the known layer-wise hierarchy in LLMs—early layers for syntax and late layers for semantics (Tenney et al., 2019; Rogers et al., 2020)—to a more fine-grained, block-internal level of computational neuroanatomy.

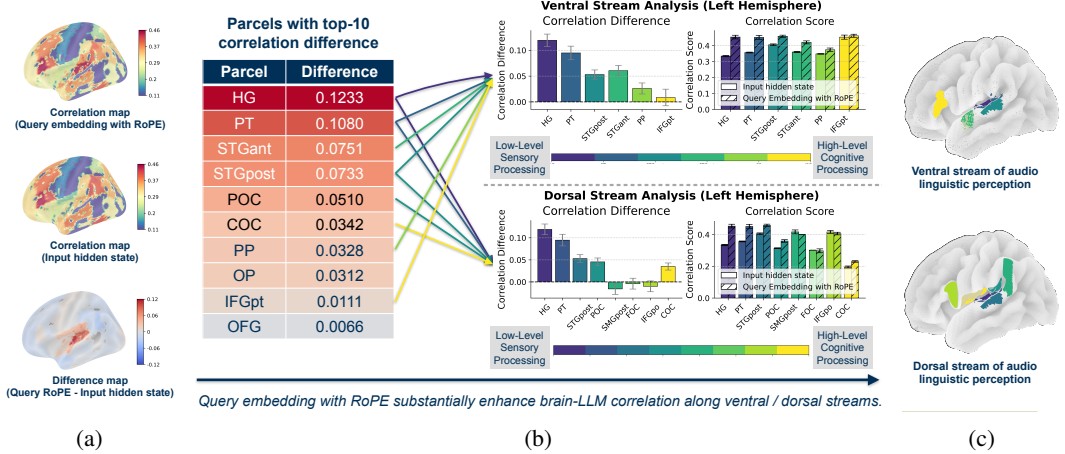

Figure 4: Per-head query with Rotary Positional Embedding (RoPE) substantially enhance LLM-brain correlation along the ventral and dorsal streams for audio linguistic perception. (a) Correlation map for Query embedding with RoPE and input hidden state. The difference map delineates both the ventral and dorsal streams. (b) Parcels with top-10 correlation difference based on Harvard-Oxford structural atlas clearly outline the ventral and dorsal streams, from low-level sensory processing to high-level cognitive processing. (c) Ventral and dorsal streams for audio linguistic perception.

## 4.2    THE ROLE OF RoPE IN DELINEATING THE BRAIN'S AUDITORY STREAMS

Among all intermediate states examined in our computational neuroanatomy analysis, the per-head query with RoPE provides the most substantial and systematic improvement in brain alignment. We focus on this specific intermediate state inside the attention mechanism and demonstrate its superior correspondence with the neuroanatomy of auditory streams in the brain.

We compare how the per-head query with RoPE improves the brain correlation from the input hidden state through detailed computational neuroanatomy. As illustrated in Figure 4a, we observe strong improvement around the Sylvian fissure in the difference map. We then rank the top-10 parcels with highest correlation difference. The ranking table in Figure 4b strikingly delineates the brain's canonical dorsal and ventral streams for auditory language processing—a fundamental principle of auditory neuroanatomy—as labeled in Figure 4c. The largest improvement is observed in the primary auditory cortex (Heschl's gyrus, HG), with the effect cascading along both anatomical streams to regions like the planum temporale (PT) and the superior temporal gyrus (STG) in Figure 4b.

This computational neuroanatomy result is significant for two reasons. First, it provides the first strong evidence of LLM-brain alignment in low-level sensory processing regions, especially in Heschl's gyrus. The prevailing consensus has been that LLM-brain alignment is primarily sensitive to high-level semantic information (Caucheteux & King, 2022; Kauf et al., 2024; Doerig et al., 2025; Goldstein et al., 2025). Our computational neuroanatomy findings suggest that this was a limitation of the representations being used, not the methodology itself; by examining the correct intermediate computational state through a neuroanatomical lens, we uncover a deep correspondence in how the model and brain align with each other when processing the fundamental auditory signal.

Second, this provides the first neurobiological evidence for the functional role of Rotary Positional Embeddings (RoPE) within a computational neuroanatomy framework. By isolating the effect of RoPE (comparing per-head query with and without it), we find that RoPE is critical for this alignment with the low-level auditory stream. The winning ratio analysis reveals a striking pattern: while per-head query without RoPE wins in only 7.82% of auditory cortex voxels, per-head query with RoPE dominates overwhelmingly at 73.88% (Appendix D)—a nearly tenfold increase. This dramatic shift is specific to auditory regions; in the language network, per-head query actually outperforms its RoPE-enhanced counterpart (19.43% vs. 9.82%, Appendix D), suggesting that RoPE's contribution is precisely tuned to the computational demands of low-level sensory processing. This regional specificity demonstrates that RoPE systematically enhances the ability of attention heads to capture the sequential and positional information critical for processing the structural nature of

speech—a function that mirrors the neuroanatomical role of the auditory ventral stream. This computational neuroanatomy discovery bridges the gap between architectural design choices in artificial systems and their biological counterparts.

# 5 MindTransformer: A Computational Neuroanatomy Alignment Framework

Our computational neuroanatomy analysis in the preceding section reveals a key insight: the most neurally-relevant information in a transformer is not localized to a single representation but is distributed across a diverse ecosystem of internal computational states that map to distinct neuroanatomical regions. This motivates us to develop **MindTransformer**, a principled framework that systematically leverages these distributed representations to achieve superior brain alignment.

## 5.1 The MindTransformer Framework

We propose MindTransformer in two complementary modes, each addressing different aspects of the LLM-brain alignment challenge:

**MindTransformer Mode 1: Optimal Single-State Selection.** In this mode, we systematically evaluate all thirteen intermediate states through independent ridge regression models to identify the single best predictor of brain activity. For each state, we train a voxel-wise encoding model in the training set and compute the prediction correlation in the testing set. The state achieving the highest correlation across voxels or ROIs is selected as the optimal representation. This approach moves beyond the arbitrary selection of hidden states or context vectors, instead letting the brain's response patterns guide the choice of representation. The per-voxel version of this mode is exactly the method implemented in Section 4 to obtain the analysis of winning ratio in Figure 1b and intra-block hierarchy in Figure 2.

**MindTransformer Mode 2: Multi-State Feature Integration.** While Mode 1 identifies the single best state, Mode 2 further integrate the information provided by different intermediate states to deliver even better prediction correlation. Different brain regions may be best explained by the combination of different computational components. We first concatenate multiple intermediate state representations to create a comprehensive feature set. We then train a ridge regression model on this high-dimensional space, where the learned weights ($\beta$) indicate each feature's importance for predicting brain activity. To balance model complexity with interpretability, we select the top-$k$ features with the largest absolute $\beta$ values (where $k$ is set to $D_{\text{model}}$) and train a final refined model. The feature selection and refined model are completed all in the training set. This two-stage approach offers dual benefits: (1) improved prediction performance through feature combination, and (2) enhanced interpretability by revealing which specific features from which states are most neurally relevant.

## 5.2 Experimental Validation

We evaluate MindTransformer against two established baselines across auditory / language cortex:

- **Standard Baseline:** An encoding model using only the input hidden state, representing the traditional approach in LLM-brain alignment studies (Antonello et al., 2023; Gokce & Schrimpf, 2025; Ren et al., 2025; Bonnasse-Gahot & Pallier, 2024; Sun & Moens, 2023; Aw et al., 2024; Oota et al., 2025; 2022; Aw & Toneva, 2023).
- **Context Vector Baseline:** An encoding model using the per-head context vector, recently shown to be effective for brain alignment (Kumar et al., 2024).

Table 1 demonstrates the substantial improvements achieved by both MindTransformer modes. Mode 1, despite using only a single optimally-selected state, shows remarkable gains over both baselines, with particularly dramatic improvements in primary auditory regions. In Heschl's Gyrus, Mode 1 achieves a correlation of 0.454, representing a 27.5% improvement over the standard baseline (0.356) and a 25.1% improvement over the context vector baseline (0.363). Mode 2 further enhances performance through multi-state integration, reaching 0.467 in Heschl's Gyrus—a 31.0% improvement over the standard baseline.

Table 1: Performance comparison of encoding models across auditory and language cortex. Values represent mean correlation (±std) computed across 21 LLMs with all transformer layers. The improvement column (Imp.) shows relative gain from Standard Baseline to MindTransformer Mode 2.

| Brain Region | Standard Baseline | Context Vector Baseline | Proposed (Mode 1) | Proposed (Mode 2) | Imp. (%) |
|---|---|---|---|---|---|
| Heschl's Gyrus | 0.356 (±0.049) | 0.363 (±0.049) | 0.454 (±0.059) | **0.467** (±0.056) | +31.0 |
| Planum Temporale | 0.341 (±0.048) | 0.333 (±0.050) | 0.418 (±0.060) | **0.436** (±0.055) | +27.8 |
| STG (anterior) | 0.367 (±0.043) | 0.351 (±0.044) | 0.419 (±0.057) | **0.435** (±0.051) | +18.7 |
| STG (posterior) | 0.423 (±0.057) | 0.407 (±0.054) | 0.462 (±0.044) | **0.477** (±0.043) | +12.6 |
| *Auditory Average* | 0.372 | 0.363 | 0.438 | **0.454** | +22.0 |
| MTG (anterior) | 0.342 (±0.072) | 0.323 (±0.067) | **0.357** (±0.070) | 0.351 (±0.066) | +2.5 |
| MTG (posterior) | 0.356 (±0.101) | 0.342 (±0.098) | **0.368** (±0.102) | 0.367 (±0.104) | +3.2 |
| MTG (temp-occipital) | 0.405 (±0.086) | 0.392 (±0.087) | **0.417** (±0.087) | 0.415 (±0.086) | +2.5 |
| IFG (pars opercularis) | 0.396 (±0.052) | 0.377 (±0.053) | **0.408** (±0.053) | 0.406 (±0.054) | +2.4 |
| IFG (pars triangularis) | 0.452 (±0.058) | 0.430 (±0.059) | 0.464 (±0.058) | **0.465** (±0.058) | +2.9 |
| Angular Gyrus | 0.403 (±0.062) | 0.385 (±0.062) | **0.419** (±0.061) | 0.409 (±0.059) | +1.4 |
| *Language Average* | 0.392 | 0.375 | **0.406** | 0.402 | +2.6 |

The contrast between auditory and language regions reveals the targeted effectiveness of our approach. While auditory regions show an average improvement of 22.0%, language network regions show more modest gains averaging 2.6%. This regional specificity validates our computational neuroanatomy hypothesis: the identification of per-head query with RoPE as the optimal state for auditory processing (as shown in Appendix D) translates directly into substantial performance gains in these low-level sensory areas. Notably, in some language regions, Mode 1 slightly outperforms Mode 2 (e.g., Angular Gyrus: 0.419 vs. 0.409), suggesting that for high-level semantic processing, a single well-chosen state may be more effective than multi-state integration.

To contextualize the magnitude of these improvements, consider that scaling LLMs from 270M to 123B parameters—a 456-fold increase in model size—typically yields correlation improvements of approximately 0.02–0.04 in auditory regions (Figure 5). In contrast, MindTransformer Mode 2 achieves improvements of 0.111 in Heschl's Gyrus and 0.095 in Planum Temporale through computational neuroanatomy insights alone, without any increase in model parameters. This demonstrates that understanding the internal computational structure of transformers can yield gains exceeding those from massive scale increases by several folds, particularly in sensory processing regions where traditional approaches have struggled.

### 5.3 PER-SUBJECT ANALYSIS

To validate that our group-averaged findings are not artifacts of averaging, we analyzed the first five subjects of the *Le Petit Prince* dataset individually on Llama 3.2 1B. [5] Consistent with group-level results, standard hidden states remain suboptimal, winning in less than 20% of voxels (Appendix Figure 7). MindTransformer reliably improves performance at the individual level: Mode 2 yields a 21.9% gain in Heschl's Gyrus ($r = 0.127$ vs $0.104$) and consistent boosts across auditory regions (Appendix Table 11). Furthermore, the intra-block hierarchy persists individually: as shown in Appendix Figure 9, early cortical regions (HG/PT) map to the block's "entry" (RoPE-query) while association areas align with deeper FFN states, confirming that the computational-cortical isomorphism is robust to individual variation.

### 5.4 ROBUSTNESS ANALYSIS WITH BASELINE ADJUSTMENT AND CONTROLLED REGRESSORS

To verify that our results are driven by the specific information content of transformer states rather than statistical confounds, we implemented two rigorous controls. First, we addressed the concern that states with higher dimensionality (e.g., FFN) yield higher correlations solely due to an increased

---

[5]The result for all the 21 LLMs are demonstrated in Appendix Table 14.

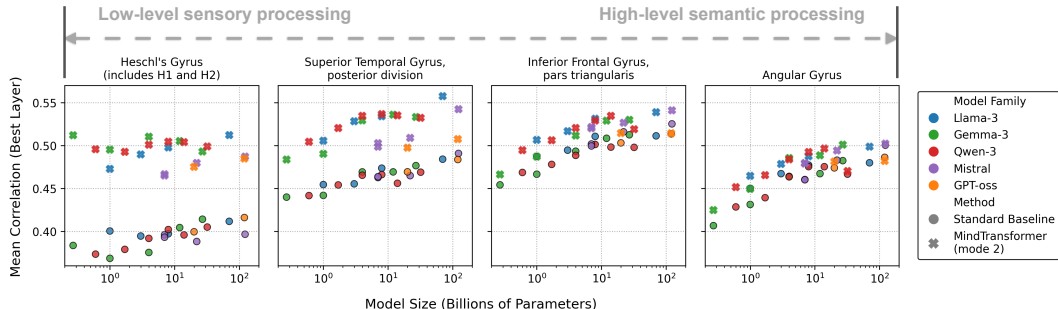

Figure 5: Correlation improvement along model sizes for 5 families of LLMs. The improvement brought by MindTransformer in low-level sensory cortex like Heschl's Gyrus is substantially larger than scaling model size by 456 times (from 0.27B to 123B).

number of regressors. By restricting all intermediate states to a fixed dimensionality ($D = 2048$) via top-$k$ feature selection, we confirmed that the original setup with slightly inflate the performance for FFN as shown in Appendix Figure 8, but the functional dissociation persists: restricted FFN states remain optimal for the language network, while lower-dimensional RoPE-enhanced states continue to dominate the auditory cortex (Appendix Figure 7 and Table 11).

Second, we benchmarked performance against random and GloVe embedding baselines (Bonnasse-Gahot & Pallier, 2024). Since fMRI signals possess inherent temporal structure tied to stimulus onsets, even random embeddings can yield non-trivial correlations by tracking these basic statistics. Adjusting for this baseline isolates the specific contribution of the LLM's contextual processing. As detailed in Appendix Tables 10–13, low-level sensory regions exhibit weak correlations with random embeddings ($r \approx 0.026$). Consequently, the baseline-adjusted improvement is substantial: Mind-Transformer outperforms the standard baseline by **29.2%** (random-adjusted) and **46.0%** (GloVe-adjusted) in Heschl's Gyrus. This confirms that the alignment is driven by the unique, context-aware structural dynamics captured by LLM, rather than generic onset tracking or static lexical features.

## 6 DISCUSSION AND CONCLUSION

This work introduces computational neuroanatomy as a systematic framework for understanding LLM-brain alignment. By dissecting the transformer block's internal computations and mapping them to precise neuroanatomical structures, we reveal that the standard practice of using single hidden states overlooks the rich, distributed neural information encoded throughout the transformer's computational pipeline. Our findings demonstrate that different intermediate states correspond to distinct brain systems with remarkable anatomical precision—from low-level sensory processing in Heschl's gyrus to high-level integration in association cortices.

The discovery that RoPE specifically enhances alignment with the brain's auditory processing streams exemplifies how computational neuroanatomy can bridge artificial and biological intelligence, providing neurobiological validation for architectural design choices. Our MindTransformer framework operationalizes these insights, achieving significant performance by intelligently combining neuroanatomically-relevant features from across the transformer block.

These results have profound implications for both neuroscience and AI. For neuroscience, computational neuroanatomy offers a new lens for understanding how the brain implements language processing, with LLMs serving as explicit computational hypotheses. For AI, our findings suggest that brain-inspired architectural modifications—guided by computational neuroanatomy—could lead to more human-like and interpretable language models.

Future work should extend this framework to Vision Transformers (ViTs) and multimodal architectures. We hypothesize that the intra-block hierarchy—early attention states aligning with sensory inputs and FFNs with semantic objects—may also characterize the visual ventral stream. Mind-Transformer's ability to dynamically integrate these distinct computational components offers a promising path for revealing universal principles of intelligence across modalities and creatures.

ACKNOWLEDGMENTS

This work was supported in part by the Wayne J. Holman Endowed Chair and the Executive Vice President for Research at Georgia Tech.

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

## A ARCHITECTURAL PARAMETERS OF VARIOUS LLM FAMILIES

Architectural parameters of various LLM families are provided in Table 2.

Table 2: Architectural Parameters of Various LLM Families.

| Family | Model Variant (Model size) | $D_{\text{model}}$ | $N_q$ | $N_{kv}$ | $D_{\text{head}}$ | $D_{\text{ffn}}$ | $N_{\text{layer}}$ |
|---|---|---|---|---|---|---|---|
| **Llama** | Llama 3.2 Instruct (1B) | 2048 | 32 | 8* | 64 | 8192 | 16 |
| | Llama 3.2 Instruct (3B) | 3072 | 24 | 8* | 128 | 8192 | 28 |
| | Llama 3.1 Instruct (8B) | 4096 | 32 | 8* | 128 | 14336 | 32 |
| | Llama 3.3 Instruct (70B) | 8192 | 64 | 8* | 128 | 28672 | 80 |
| **Qwen** | Qwen3 (0.6B) | 1024 | 16 | 8* | 128 | 3072 | 28 |
| | Qwen3 (1.7B) | 2048 | 16 | 8* | 128 | 6144 | 28 |
| | Qwen3 (4B) | 2560 | 32 | 8* | 128 | 9728 | 36 |
| | Qwen3 (8B) | 4096 | 32 | 8* | 128 | 12288 | 36 |
| | Qwen3 (14B) | 5120 | 40 | 8* | 128 | 17408 | 40 |
| | Qwen3 (32B) | 5120 | 64 | 8* | 128 | 25600 | 64 |
| **Mistral** | Mistral 7B Instruct v0.2 (7B) | 4096 | 32 | 8* | 128 | 14336 | 32 |
| | Mistral 7B Instruct v0.3 (7B) | 4096 | 32 | 8* | 128 | 14336 | 32 |
| | Mistral Small Instruct (22B) | 6144 | 48 | 8* | 128 | 16384 | 56 |
| | Mistral Large Instruct (123B) | 12288 | 96 | 8* | 128 | 28672 | 88 |
| **GPT** | GPT-oss (20B) | 2880 | 64 | 8* | 64 | 2880† | 24 |
| | GPT-oss (120B) | 2880 | 64 | 8* | 64 | 2880† | 36 |
| **Gemma** | Gemma 3 Instruct (270M) | 640 | 4 | 1* | 256 | 2048 | 18 |
| | Gemma 3 Instruct (1B) | 1152 | 4 | 1* | 256 | 6912 | 26 |

**Table 2 – continued from previous page**

| Family | Model Variant | $D_{\text{model}}$ | $N_q$ | $N_{kv}$ | $D_{\text{head}}$ | $D_{\text{ffn}}$ | $N_{\text{layer}}$ |
|---|---|---|---|---|---|---|---|
| | Gemma 3 Instruct (4B) | 2560 | 8 | 4* | 256 | 10240 | 34 |
| | Gemma 3 Instruct (12B) | 3840 | 16 | 8* | 256 | 15360 | 48 |
| | Gemma 3 Instruct (27B) | 5376 | 32 | 16* | 128 | 21504 | 62 |

\* Model uses GQA or MQA, where $N_{kv} < N_q$.

† Value is per expert in a Mixture-of-Experts (MoE) model.

## B    TRANSFORMER STATES AND TENSOR SHAPES

Transformer states and tensor shapes are provided in Table 3.

Table 3: Complete list of intermediate states extracted from transformer blocks with their dimensions and descriptions

| Stage | State Name | Tensor Shape | Description |
|---|---|---|---|
| **Block Input** | Input Hidden State | $(B, S, D_{\text{model}})$ | Output from the previous block |
| | Pre-Attention Normalized State | $(B, S, D_{\text{model}})$ | Output of the first LayerNorm |
| **Attention Mechanism** | Per-Head Query (Q) | $(B, N_q, S, D_{\text{head}})$ | Q projection output |
| | Per-Head Key (K) | $(B, N_{kv}, S, D_{\text{head}})$ | K projection output |
| | Per-Head Q with RoPE | $(B, N_q, S, D_{\text{head}})$ | Q projection output with RoPE applied |
| | Per-Head K with RoPE | $(B, N_{kv}, S, D_{\text{head}})$ | K projection output with RoPE applied |
| | Per-Head Value (V) | $(B, N_{kv}, S, D_{\text{head}})$ | V projection output |
| | Per-Head Context Vector | $(B, N_q, S, D_{\text{head}})$ | Context vector after attention mechanism |
| | Combined Attention Output | $(B, S, D_{\text{model}})$ | Output of the final attention projection |
| **FFN & Residuals** | Post-Attention Hidden State | $(B, S, D_{\text{model}})$ | Sum from the first residual connection |
| | Pre-FFN Normalized State | $(B, S, D_{\text{model}})$ | Output of the second LayerNorm |
| | FFN Activated State | $(B, S, D_{\text{ffn}})$ | Output of FFN up-projection |
| | FFN Output | $(B, S, D_{\text{model}})$ | Output of FFN down-projection |

## C    REGIONS OF INTEREST DEFINITION AND CHARACTERISTICS

Regions of interest definition and characteristics are provided in Table 4.

Table 4: Anatomical and functional characteristics of regions of interest (ROIs) used in our analysis

| ROI | Voxel Count | Anatomical Coverage | Constituent Parcels | Functional Significance |
|---|---|---|---|---|
| **Whole-Brain Voxels** | 25,870 | Complete cortical and subcortical coverage within the fMRI acquisition field of view | All 48 cortical regions from the Harvard-Oxford atlas plus subcortical structures | Comprehensive neural processing, serving as a baseline for all sensory, motor, and higher-order association cortices |
| **Fedorenko Language Network** | 1,740 | Distributed, strongly left-lateralized fronto-temporal-parietal regions defined by language-selective functional localizers | Orbital inferior frontal (IFGorb) Inferior frontal gyrus (IFG) Middle frontal gyrus (MFG) Anterior temporal (AntTemp) Posterior temporal (PostTemp) Angular gyrus (AngGyr) | High-level language comprehension, including semantic processing, syntactic parsing, discourse integration, and abstract linguistic reasoning |
| **Auditory-Sensory Cortex** | 325 | Bilateral superior temporal regions surrounding the Sylvian fissure, including primary and secondary auditory cortex | Heschl's Gyrus (HG) Planum Temporale (PT) Superior Temporal Gyrus, anterior division (STGpost) Superior Temporal Gyrus, posterior division (STGant) | Low-level acoustic feature extraction, spectrotemporal analysis, pitch processing, and early stages of speech perception |

The three ROIs represent a hierarchical organization of language processing in the brain. The whole-brain ROI (25,870 voxels) provides an unbiased view of all cortical and subcortical processing. The Fedorenko language network (1,740 voxels) (Fedorenko et al., 2010) represents domain-specific, high-level language regions identified through functional localizers, showing selective responses to linguistic versus non-linguistic stimuli. The auditory-sensory cortex (325 voxels) captures early, stimulus-driven processing of acoustic and phonological features critical for speech perception. This hierarchical organization allows us to trace how different transformer components align with the progression from sensory to semantic processing in the human brain.

Table 5: Performance comparison across major brain networks. Values represent mean correlation (± standard deviation across models).

| Brain Region | Standard Baseline | Context Vector Baseline | Optimal State |
|---|---|---|---|
| Whole Brain | 0.275 (±0.011) | 0.280 (±0.014) | 0.296 (±0.011) |
| Language Network | 0.433 (±0.016) | 0.429 (±0.019) | 0.450 (±0.015) |
| Audio Cortex | 0.407 (±0.010) | 0.416 (±0.021) | 0.475 (±0.015) |

## D   STATE-WISE WINNING RATIOS ACROSS BRAIN REGIONS

To provide a comprehensive view of how different intermediate states dominate across various brain regions, we analyzed the winning ratios (percentage of voxels where each state provides the best encoding performance) for three distinct regions: whole-brain, auditory-sensory cortex, and Fedorenko language network, in Figure 6.

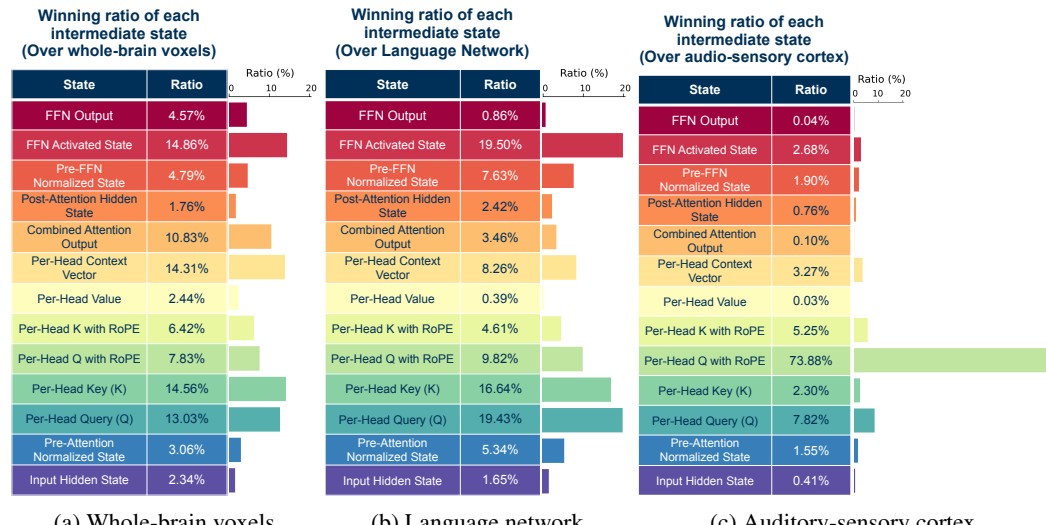

(a) Whole-brain voxels     (b) Language network     (c) Auditory-sensory cortex

Figure 6: Winning ratio distributions of intermediate states across different brain regions. The dominance of different states varies dramatically by region, with FFN states being more prominent in language networks and per-head query with RoPE dominating in auditory cortex.

Tables 6, 7, and 8 provide detailed statistics of winning ratio across all five model families.

Table 6: Per-LLM-family winning ratio for whole-brain voxel (%)

| State | Mistral | GPT-oss | Llama-3 | Qwen-3 | Gemma-3 |
|---|---|---|---|---|---|
| Input Hidden State | 1.22 | 2.54 | 1.40 | 0.66 | 2.61 |
| Pre-Attn Norm State | 2.36 | 8.17 | 3.46 | 2.40 | 2.03 |
| Per-Head Q | 13.92 | 10.55 | 12.10 | 14.18 | 12.69 |
| Per-Head K | 13.18 | 8.01 | 14.03 | **16.14** | 16.82 |
| Per-Head Q w/ RoPE | 6.16 | 4.48 | 6.87 | 8.31 | 10.68 |
| Per-Head K w/ RoPE | 3.85 | 4.12 | 5.13 | 9.88 | 6.27 |
| Per-Head V | 2.19 | 5.49 | 2.18 | 2.23 | 1.88 |
| Per-Head Context | **17.80** | **22.98** | 12.85 | 12.82 | 11.01 |
| Combined Attn Output | 13.09 | 9.82 | 13.83 | 11.51 | 6.19 |
| Post-Attn Hidden State | 1.38 | 5.94 | 2.93 | 1.30 | 0.00 |
| Pre-FFN Norm State | 2.92 | 13.78 | 3.79 | 3.37 | 5.20 |
| FFN Activated State | 16.98 | 0.00 | **15.60** | 13.31 | **20.36** |
| FFN Output | 4.96 | 4.12 | 5.81 | 3.88 | 4.26 |

Table 7: Per-LLM-family winning ratio for Fedorenko language network (%)

| State | Mistral | GPT-oss | Llama-3 | Qwen-3 | Gemma-3 |
|---|---|---|---|---|---|
| Input Hidden State | 1.77 | 1.35 | 1.70 | 0.37 | 3.16 |
| Pre-Attn Norm State | 4.90 | 14.89 | 5.93 | 3.48 | 3.64 |
| Per-Head Q | **24.91** | 12.13 | 17.56 | **24.04** | 13.91 |
| Per-Head K | 15.47 | 2.39 | 15.63 | 18.63 | 21.70 |
| Per-Head Q w/ RoPE | 6.18 | 5.14 | 7.83 | 13.97 | 11.22 |
| Per-Head K w/ RoPE | 1.75 | 1.58 | 7.43 | 7.08 | 2.87 |
| Per-Head V | 0.14 | 1.26 | 0.09 | 0.41 | 0.44 |
| Per-Head Context | 12.30 | 16.87 | 6.18 | 6.07 | 5.87 |
| Combined Attn Output | 4.34 | 1.24 | 3.82 | 3.76 | 2.99 |
| Post-Attn Hidden State | 1.55 | 9.86 | 4.81 | 0.95 | 0.00 |
| Pre-FFN Norm State | 5.46 | **32.39** | 8.64 | 4.25 | 2.72 |
| **FFN Activated State** | 20.07 | 0.00 | **19.07** | 16.59 | **30.68** |
| FFN Output | 1.15 | 0.92 | 1.32 | 0.39 | 0.79 |

Table 8: Per-LLM-family winning ratio for auditory-sensory cortex (%)

| State | Mistral | GPT-oss | Llama-3 | Qwen-3 | Gemma-3 |
|---|---|---|---|---|---|
| Input Hidden State | 0.62 | 0.15 | 0.31 | 0.05 | 0.86 |
| Pre-Attn Norm State | 2.00 | 3.85 | 1.69 | 0.36 | 1.60 |
| Per-Head Q | 17.38 | 7.69 | 7.62 | 5.28 | 3.45 |
| Per-Head K | 3.38 | 0.00 | 0.85 | 2.67 | 3.08 |
| **Per-Head Q w/ RoPE** | **66.77** | **42.15** | **72.85** | **84.31** | **80.55** |
| Per-Head K w/ RoPE | 4.38 | 0.00 | 10.54 | 4.26 | 4.98 |
| Per-Head V | 0.00 | 0.00 | 0.00 | 0.10 | 0.00 |
| Per-Head Context | 1.23 | 29.85 | 0.08 | 0.31 | 0.37 |
| Combined Attn Output | 0.00 | 0.00 | 0.00 | 0.10 | 0.31 |
| Post-Attn Hidden State | 0.92 | 4.31 | 0.69 | 0.15 | 0.00 |
| Pre-FFN Norm State | 1.15 | 11.85 | 2.23 | 0.31 | 0.18 |
| FFN Activated State | 2.15 | 0.00 | 3.15 | 2.10 | 4.49 |
| FFN Output | 0.00 | 0.15 | 0.00 | 0.00 | 0.12 |

The stark differences in winning ratios across brain regions reveal distinct computational preferences. In the language network (Table 7) shows a more distributed pattern with FFN Activated State and Per-Head Q/K states sharing dominance, reflecting the complex, multi-faceted nature of language processing. In contrast, the auditory-sensory cortex (Table 8), Per-Head Q with RoPE dominates overwhelmingly, winning in 66.77–84.31% of voxels across model families, highlighting its critical role in low-level sensory processing. These regional specializations provide strong evidence for our computational neuroanatomy framework, demonstrating that different transformer components align with functionally distinct brain systems.

## E    QUANTIFICATION OF COMPUTATIONAL DEPTH

To quantitatively compare the processing hierarchy of the biological brain with the internal computational flow of the transformer block, we introduce a method to map discrete transformer states onto a continuous hierarchical axis. We utilize a consistent terminology where **Computational Depth** refers to the normalized position of a specific intermediate state, and **Weighted Computational Depth** refers to the aggregate metric for a brain region.

**State Selection and Normalization.**    While we extract 13 intermediate states for general analysis, for the specific purpose of quantifying the intra-block hierarchy, we define a focused subset of states $\mathcal{S}$ that represents the core computational trajectory. We identify the *Per-Head Query* as the functional entry point for the hierarchy, as it is the first layer consistently exhibiting alignment with early sensory cortices. We exclude the initial input and pre-attention normalization states as they precede this functional entry point. Additionally, we exclude Key and Value states to avoid redundancy, as they occupy the same topological layer as Queries but exhibit weaker alignment.

The resulting ordered set $\mathcal{S}$ consists of the following 8 states:

1. Per-head Query
2. Per-head Query w/ RoPE
3. Per-head Context Vector
4. Combined Attention Output
5. Post-Attention Hidden State
6. Pre-FFN Normalized State
7. FFN Activated State
8. FFN Output

We assign a normalized **Computational Depth** score $\delta(s_i) \in [0, 1]$ to each state $s_i \in \mathcal{S}$. The depth is calculated as:

$$\delta(s_i) = \frac{i - 1}{|\mathcal{S}| - 1} \tag{1}$$

where $|\mathcal{S}| = 8$. This establishes a linear coordinate system where the Per-head Query corresponds to $\delta = 0$ and the FFN Output corresponds to $\delta = 1$.

**Weighted Computational Depth.** To map brain regions onto this axis, we compute the **Weighted Computational Depth** $\bar{D}_\mathcal{R}$ for a given Region of Interest (ROI) $\mathcal{R}$. For each voxel $v$ within the region, we first identify the optimal state $s^*(v)$ from the set $\mathcal{S}$ that yields the highest encoding correlation. We then compute the average depth across all voxels in the region:

$$\bar{D}_\mathcal{R} = \frac{1}{|V_\mathcal{R}|} \sum_{v \in V_\mathcal{R}} \delta(s^*(v)) \tag{2}$$

This metric represents the "center of gravity" of alignment for a brain region within the transformer block's computational order. In Figure 3, we correlate this metric with the cortical hierarchy of brain regions along the audio stream and the language network.

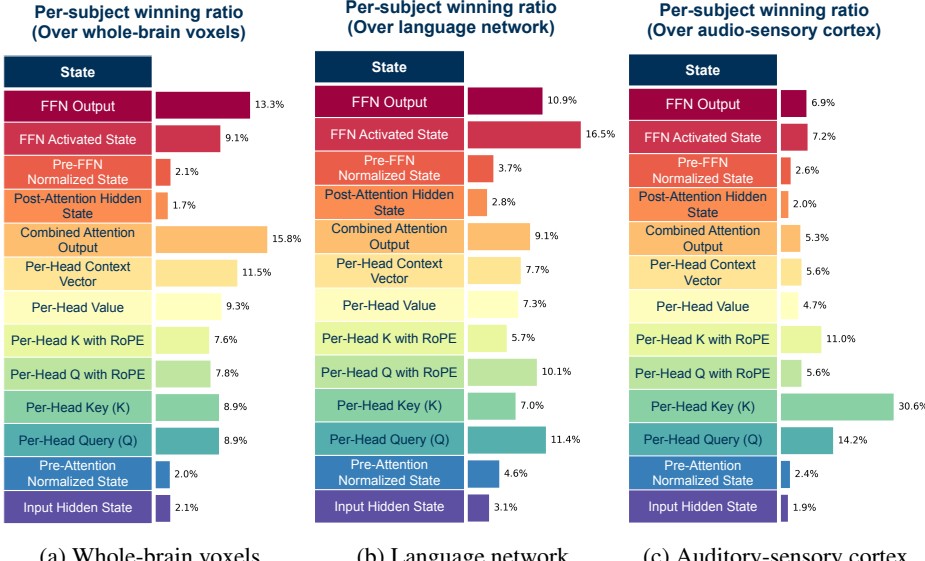

(a) Whole-brain voxels      (b) Language network      (c) Auditory-sensory cortex

Figure 7: Per-subject winning ratio distributions of intermediate states across different brain regions over Llama 3.2 1B. The dominance of different states varies dramatically by region, with FFN states being more prominent in language networks and per-head query with RoPE dominating in auditory cortex.

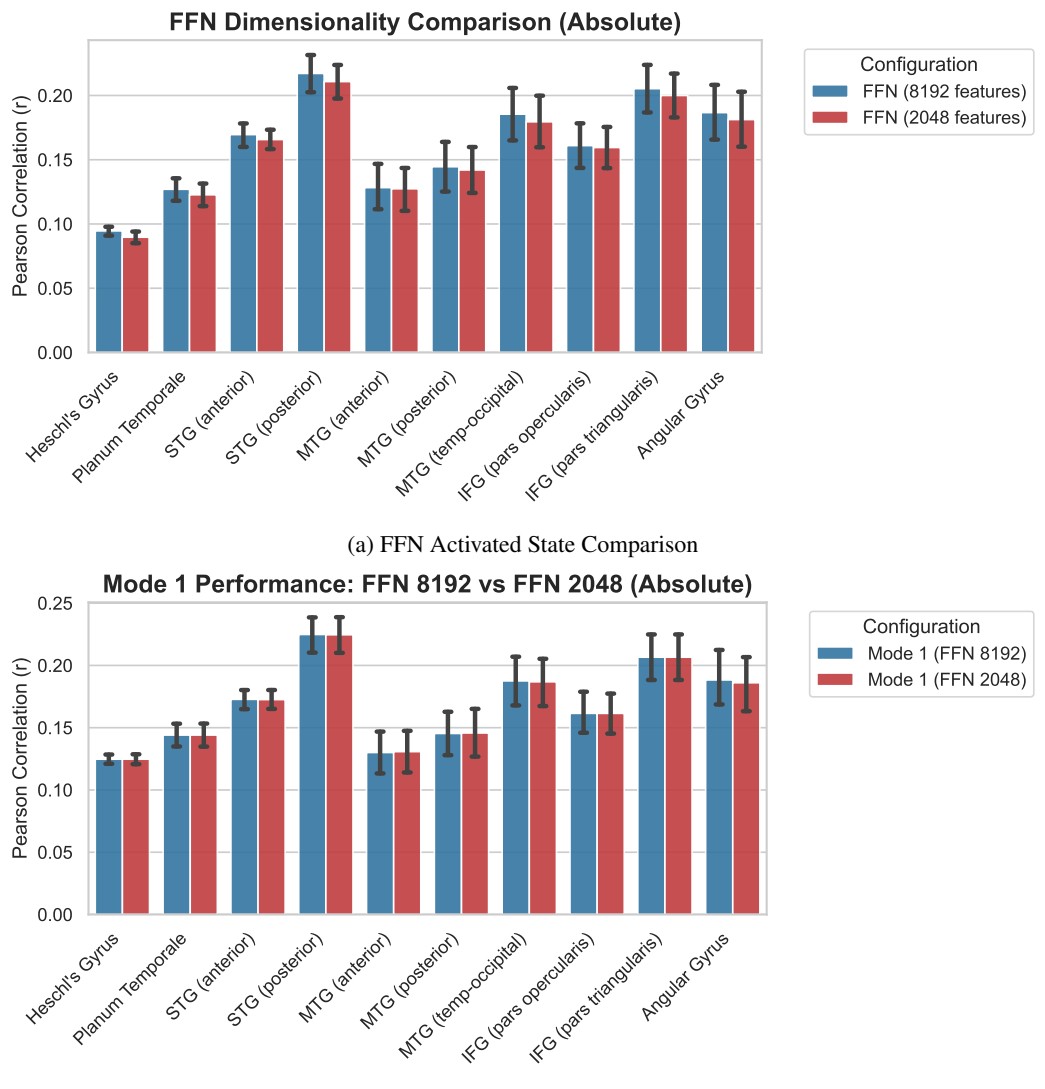

(a) FFN Activated State Comparison

(b) MindTransformer Mode 1 Comparison

Figure 8: Impact of dimensionality control on alignment performance (Llama-3.2 1B). (a) Restricting the FFN activated state from its native dimensionality (8192) to a controlled size (2048) via top-$k$ feature selection results in a minor relative performance degradation no more than 5.19%, indicating that the high feature count is not the primary driver of alignment. (b) For MindTransformer Mode 1, the degradation is negligible (<1.19%), demonstrating the framework's robustness: even when FFN performance dips slightly, the optimal selection from the remaining 12 states maintains high alignment accuracy.

Table 9: Per-subject performance comparison across major brain networks. Values represent mean correlation ($\pm$ for standard deviation across 5 subjects).

| Brain Region | Standard Baseline | Context Vector Baseline | Optimal State |
|---|---|---|---|
| Whole Brain | 0.098 ($\pm$0.019) | 0.103 ($\pm$0.018) | 0.119 ($\pm$0.019) |
| Language Network | 0.187 ($\pm$0.034) | 0.184 ($\pm$0.030) | 0.202 ($\pm$0.034) |
| Audio Cortex | 0.162 ($\pm$0.019) | 0.163 ($\pm$0.019) | 0.189 ($\pm$0.022) |

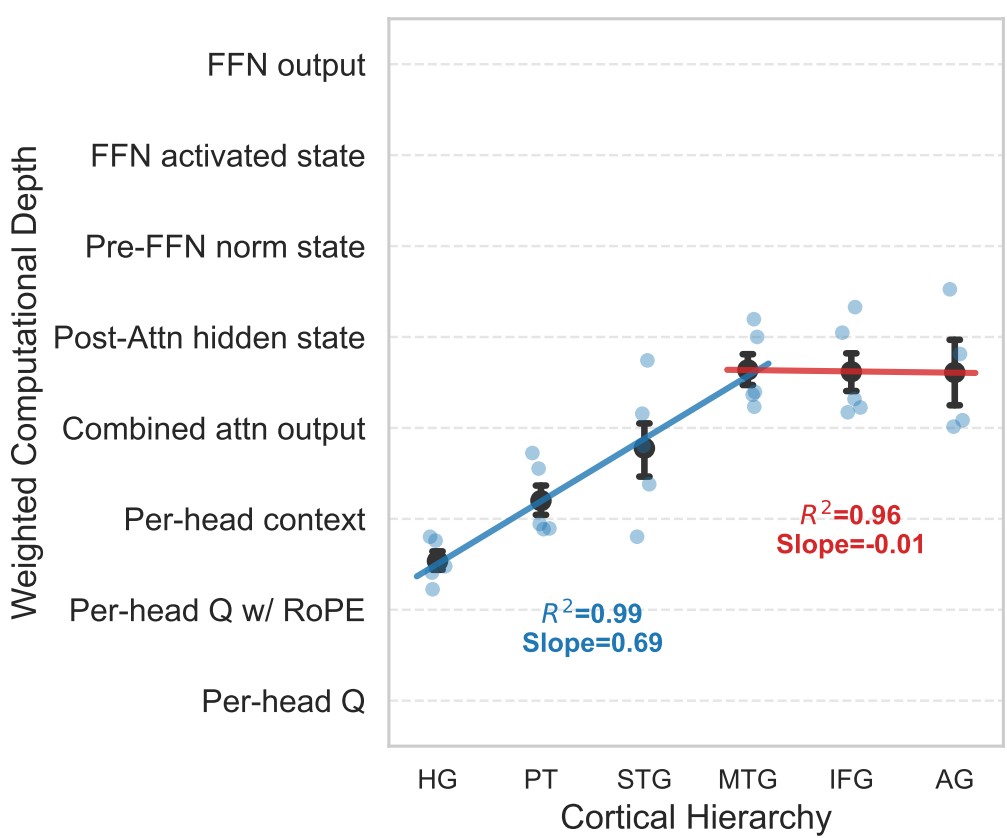

Figure 9: **Per-subject analysis of Weighted Computational Depth versus Cortical Depth (Llama-3.2 1B).** We reproduce the hierarchy analysis for the first five subjects individually. Consistent with the group-level observations (Figure 3), each subject exhibits a robust topological alignment: early auditory regions (HG, PT) map to the "entry" layers of the transformer block (specifically per-head query with RoPE), while higher-order association areas (IFG, Angular Gyrus) align with deeper FFN states. The characteristic steep slope in the auditory stream followed by a plateau in the language network is preserved across individuals, confirming that this computational-cortical isomorphism is not an artifact of group averaging.

Table 10: Per-subject baseline correlation comparison across auditory and language regions. Values represent mean correlation (±standard deviation across the mean correlation of 5 subjects).

| Brain Region | Random Baseline | Glove Baseline |
|---|---|---|
| *Auditory Cortex* | | |
| Heschl's Gyrus | 0.026 (±0.025) | 0.054 (±0.027) |
| Planum Temporale | 0.043 (±0.026) | 0.068 (±0.027) |
| STG (anterior) | 0.051 (±0.027) | 0.092 (±0.025) |
| STG (posterior) | 0.059 (±0.044) | 0.119 (±0.041) |
| *Auditory Average* | 0.045 | 0.083 |
| *Language Network* | | |
| MTG (anterior) | 0.032 (±0.022) | 0.067 (±0.032) |
| MTG (posterior) | 0.039 (±0.027) | 0.079 (±0.039) |
| MTG (temp-occipital) | 0.053 (±0.037) | 0.117 (±0.034) |
| IFG (pars opercularis) | 0.046 (±0.027) | 0.086 (±0.037) |
| IFG (pars triangularis) | 0.061 (±0.031) | 0.123 (±0.042) |
| Angular Gyrus | 0.058 (±0.038) | 0.101 (±0.041) |
| *Language Average* | 0.048 | 0.096 |

Table 11: Per-subject raw correlation performance comparison of encoding models across auditory and language regions. Values represent mean correlation (±standard deviation across the mean correlation of 5 subjects). The improvement column (Imp.) shows the relative gain from the Standard Baseline to the proposed method (Mode 2 for auditory regions, Mode 1 for language regions).

| Brain Region | Standard Baseline | Context Vector Baseline | Proposed (Mode 1) | Proposed (Mode 2) | Imp. (%) |
|---|---|---|---|---|---|
| *Auditory Cortex* | | | | | |
| Heschl's Gyrus | 0.104 (±0.012) | 0.096 (±0.013) | 0.125 (±0.009) | **0.127** (±0.015) | +21.9 |
| Planum Temporale | 0.131 (±0.024) | 0.124 (±0.024) | 0.144 (±0.021) | **0.151** (±0.024) | +15.1 |
| STG (anterior) | 0.169 (±0.021) | 0.160 (±0.020) | 0.173 (±0.020) | **0.181** (±0.022) | +6.6 |
| STG (posterior) | 0.218 (±0.039) | 0.209 (±0.036) | 0.225 (±0.036) | **0.228** (±0.031) | +4.9 |
| *Auditory Average* | 0.156 | 0.147 | 0.167 | **0.172** | +12.1 |
| *Language Network* | | | | | |
| MTG (anterior) | 0.126 (±0.044) | 0.121 (±0.042) | **0.130** (±0.044) | 0.128 (±0.042) | +3.0 |
| MTG (posterior) | 0.143 (±0.044) | 0.138 (±0.042) | **0.145** (±0.044) | 0.143 (±0.043) | +1.5 |
| MTG (temp-occipital) | 0.184 (±0.047) | 0.176 (±0.044) | **0.188** (±0.047) | 0.182 (±0.047) | +1.8 |
| IFG (pars opercularis) | 0.158 (±0.039) | 0.150 (±0.036) | **0.162** (±0.041) | 0.159 (±0.043) | +1.9 |
| IFG (pars triangularis) | 0.206 (±0.044) | 0.190 (±0.039) | **0.207** (±0.044) | **0.207** (±0.043) | +0.3 |
| Angular Gyrus | 0.183 (±0.053) | 0.176 (±0.048) | 0.188 (±0.054) | **0.199** (±0.044) | +2.8 |
| *Language Average* | 0.167 | 0.159 | **0.170** | **0.170** | +1.9 |

Table 12: Per-subject random-embedding-adjusted performance comparison of encoding models across auditory and language regions. Values represent mean correlation (±standard deviation across the mean correlation of 5 subjects). The improvement column (Imp.) shows the relative gain from the Standard Baseline to the proposed method (Mode 2 for auditory regions, Mode 1 for language regions).

| Brain Region | Standard Baseline | Context Vector Baseline | Proposed (Mode 1) | Proposed (Mode 2) | Imp. (%) |
|---|---|---|---|---|---|
| *Auditory Cortex* | | | | | |
| Heschl's Gyrus | 0.078 (±0.016) | 0.070 (±0.017) | 0.099 (±0.016) | **0.101** (±0.019) | +29.2 |
| Planum Temporale | 0.088 (±0.022) | 0.081 (±0.018) | 0.101 (±0.017) | **0.108** (±0.017) | +22.4 |
| STG (anterior) | 0.118 (±0.019) | 0.109 (±0.020) | 0.121 (±0.018) | **0.132** (±0.023) | +12.0 |
| STG (posterior) | 0.159 (±0.029) | 0.150 (±0.027) | 0.166 (±0.021) | **0.169** (±0.028) | +6.7 |
| *Auditory Average* | 0.111 | 0.103 | 0.122 | **0.128** | +17.6 |
| *Language Network* | | | | | |
| MTG (anterior) | 0.095 (±0.023) | 0.089 (±0.021) | **0.098** (±0.023) | 0.096 (±0.021) | +4.0 |
| MTG (posterior) | 0.104 (±0.023) | 0.099 (±0.020) | **0.106** (±0.021) | 0.103 (±0.019) | +2.1 |
| MTG (temp-occipital) | 0.131 (±0.027) | 0.123 (±0.024) | **0.135** (±0.025) | 0.129 (±0.028) | +2.6 |
| IFG (pars opercularis) | 0.112 (±0.045) | 0.103 (±0.041) | **0.115** (±0.045) | 0.113 (±0.047) | +2.7 |
| IFG (pars triangularis) | 0.145 (±0.020) | 0.129 (±0.017) | **0.146** (±0.019) | 0.146 (±0.017) | +0.4 |
| Angular Gyrus | 0.125 (±0.031) | 0.118 (±0.027) | **0.130** (±0.030) | 0.128 (±0.032) | +4.1 |
| *Language Average* | 0.119 | 0.110 | **0.122** | 0.119 | +2.6 |

Table 13: Per-subject GloVe-embedding-adjusted performance comparison of encoding models across auditory and language regions. Values represent mean correlation (±standard deviation across the mean correlation of 5 subjects). The improvement column (Imp.) shows the relative gain from the Standard Baseline to the proposed method (Mode 2 for auditory regions, Mode 1 for language regions).

| Brain Region | Standard Baseline | Context Vector Baseline | Proposed (Mode 1) | Proposed (Mode 2) | Imp. (%) |
|---|---|---|---|---|---|
| *Auditory Cortex* | | | | | |
| Heschl's Gyrus | 0.049 (±0.018) | 0.041 (±0.020) | 0.070 (±0.019) | **0.072** (±0.024) | +46.0 |
| Planum Temporale | 0.063 (±0.012) | 0.056 (±0.009) | 0.076 (±0.013) | **0.082** (±0.014) | +31.4 |
| STG (anterior) | 0.077 (±0.010) | 0.068 (±0.012) | 0.080 (±0.012) | **0.085** (±0.016) | +9.8 |
| STG (posterior) | 0.099 (±0.007) | 0.090 (±0.008) | 0.106 (±0.013) | **0.109** (±0.012) | +10.7 |
| *Auditory Average* | 0.072 | 0.064 | 0.083 | **0.087** | +24.5 |
| *Language Network* | | | | | |
| MTG (anterior) | 0.059 (±0.015) | 0.054 (±0.017) | **0.063** (±0.018) | 0.061 (±0.017) | +6.3 |
| MTG (posterior) | 0.065 (±0.011) | 0.059 (±0.009) | **0.067** (±0.013) | 0.064 (±0.014) | +3.4 |
| MTG (temp-occipital) | 0.067 (±0.022) | 0.059 (±0.018) | **0.071** (±0.020) | 0.065 (±0.021) | +5.0 |
| IFG (pars opercularis) | 0.072 (±0.012) | 0.064 (±0.008) | **0.075** (±0.013) | 0.073 (±0.016) | +4.2 |
| IFG (pars triangularis) | 0.083 (±0.017) | 0.067 (±0.017) | **0.083** (±0.018) | 0.083 (±0.019) | +0.7 |
| Angular Gyrus | 0.082 (±0.016) | 0.075 (±0.010) | **0.087** (±0.018) | 0.085 (±0.016) | +6.2 |
| *Language Average* | 0.071 | 0.063 | **0.074** | 0.072 | +4.3 |

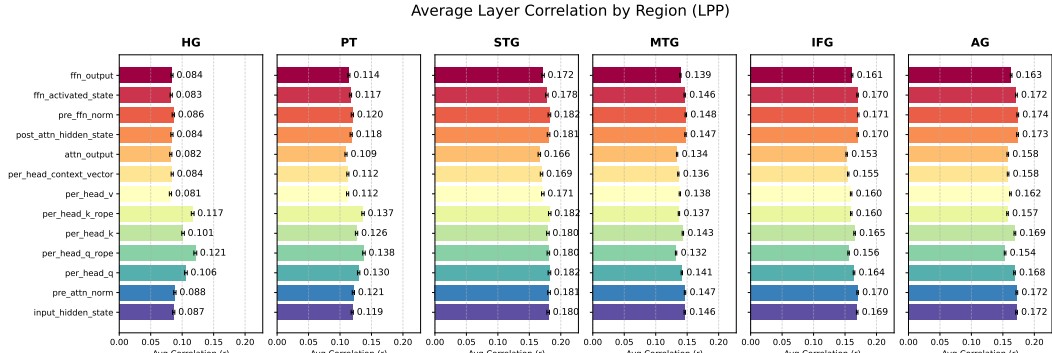

Figure 10: **Average performance of intermediate states across the cortical hierarchy.** Each subplot displays the mean Pearson correlation of the 13 intermediate states (averaged across all layers of Llama 3.2 1B) for a specific ROI. The results reveal a clear computational crossover: (1) In early auditory regions (**HG, PT**), attention-related states—specifically *Per-head Query with RoPE*—achieve the highest alignment, exhibiting a **large relative improvement** over the standard input hidden state. (2) As information progresses to the language network (**IFG, AG**), the *FFN Activated State* becomes the dominant predictor, though the relative performance delta here is more **modest**. This quantitative double dissociation validates the intra-block hierarchy proposed in our computational neuroanatomy framework.

Table 14: Per-subject encoding performance averaged over 21 LLMs across auditory cortex regions using three metrics: raw correlation, random-embedding adjusted, and GloVe adjusted. Values represent mean correlation (±standard deviation across 5 subjects). The improvement column (Imp.) shows the relative gain from the Standard Baseline to MindTransformer (Mode 2).

| Brain Region | Standard Baseline | MindTransformer (Mode 2) | Imp. (%) |
|---|---|---|---|
| *Metric 1: Raw Correlation* | | | |
| Heschl's Gyrus | 0.110 (±0.016) | **0.135** (±0.020) | +22.7 |
| Planum Temporale | 0.135 (±0.025) | **0.155** (±0.028) | +14.8 |
| STG (anterior) | 0.175 (±0.024) | **0.185** (±0.028) | +5.7 |
| STG (posterior) | 0.225 (±0.040) | **0.234** (±0.040) | +4.0 |
| *Metric 2: Random-Embedding Adjusted* | | | |
| Heschl's Gyrus | 0.084 (±0.015) | **0.110** (±0.018) | +30.9 |
| Planum Temporale | 0.092 (±0.023) | **0.112** (±0.018) | +21.7 |
| STG (anterior) | 0.123 (±0.019) | **0.136** (±0.024) | +10.5 |
| STG (posterior) | 0.166 (±0.027) | **0.175** (±0.028) | +5.4 |
| *Metric 3: GloVe-Embedding Adjusted* | | | |
| Heschl's Gyrus | 0.055 (±0.014) | **0.081** (±0.022) | +47.2 |
| Planum Temporale | 0.067 (±0.009) | **0.087** (±0.008) | +29.8 |
| STG (anterior) | 0.082 (±0.013) | **0.089** (±0.013) | +8.5 |
| STG (posterior) | 0.106 (±0.013) | **0.115** (±0.014) | +8.4 |

Table 15: Bootstrap significance analysis (FDR $q < 0.05$) comparing the Standard Baseline and MindTransformer Mode 2. Values represent the percentage of significant voxels in the region, with the average correlation ($r$) of those significant voxels in parentheses. Best performing coverage is bolded.

| Brain Region | Standard Baseline | Proposed (Mode 2) |
|---|---|---|
| *Auditory Cortex* | | |
| Heschl's Gyrus (includes H1 and H2) | 78.6% ($r = 0.141$) | **89.0%** ($r = 0.163$) |
| Planum Temporale | 88.0% ($r = 0.162$) | **93.0%** ($r = 0.165$) |
| STG (posterior) | 97.4% ($r = 0.227$) | **98.4%** ($r = 0.236$) |
| STG (anterior) | 94.0% ($r = 0.191$) | **95.7%** ($r = 0.187$) |
| *Language Network* | | |
| MTG (temporooccipital) | 90.6% ($r = 0.211$) | **94.9%** ($r = 0.199$) |
| MTG (posterior) | 78.9% ($r = 0.192$) | **82.4%** ($r = 0.170$) |
| MTG (anterior) | 82.5% ($r = 0.158$) | **86.2%** ($r = 0.152$) |
| IFG (pars opercularis) | **90.6%** ($r = 0.183$) | 89.9% ($r = 0.180$) |
| IFG (pars triangularis) | 96.8% ($r = 0.219$) | **98.5%** ($r = 0.212$) |
| Angular Gyrus | 94.0% ($r = 0.197$) | **97.3%** ($r = 0.196$) |

