# OpenReview forum: "The Mind's Transformer: Computational Neuroanatomy of LLM-Brain Alignment"
_ICLR.cc/2026/Conference — ICLR 2026 Poster_

### Official Review · Reviewer_RBfX · 2025-10-30

**Soundness:** 1
**Presentation:** 3
**Contribution:** 1
**Rating:** 2
**Confidence:** 4

**Summary:**

This paper investigates how well each component of a transformer block (e.g., "per-head query", "per-head query with RoPE", "FFN Activated State", "FFN Output") provides representations to predict brain activity during natural language processing. The work systematically compares these components across a large variety of model sizes and model families. It finally shows that combining them provides better brain scores than simply using the commonly used hidden states.

**Strengths:**

- The work investigates a large number of models from five model families, with sizes ranging from 270M to 123B parameters. The two largest models, above 100B parameters, are relatively rare in this type of literature, and not easily computationally accessible for many researchers, which could be of interest to the community.

- Another strength is the systematic approach the authors propose in studying all the different components of each transformer block.

**Weaknesses:**

There are several significant limitations that lead me to reject the paper.

- The work compares different representations that do not have the same number of features, which leads to unfair comparisons. Indeed, all else being equal, the brain predictivity increases with the number of potential regressors. Here, all the components in a transformer block do not have the same number of features, and this can vary considerably: in particular, the hidden size of the MLP block (called "ffn activation state" here) is usually much larger than the hidden size (e.g., 3x for Qwen3 0.6B, 4x for LLama 3.2 1B, or even 5x for Qwen3 32B). Unsurprisingly, it is indeed this particular component that wins most of the times (see Fig 1 and Fig 5b). One solution to provide fair comparisons might be to equate the dimensions of all the vectors, either by applying a PCA or by randomly dropping some features. For the same reason, the fact that combining all the different features (which the authors call the "Mind's Transformer") leads to better results is not very surprising nor insightful at this stage. Moreover, previous work has investigated proper ways to combine various features: see Dupré la Tour et al. (2022).

- Another important limitation of the work is the absence of proper baselines. Previous work (see e.g. Schrimpf et al., 2021; Pasquiou et al., 2022; Bonnasse-Gahot & Pallier, 2024) has shown that random baselines or untrained models can achieve pretty high brain scores. Here, it is necessary to compare the results with proper baselines such as random vectors, random embeddings, or untrained networks. For instance, in the auditory cortex, simply having random embeddings that can be used to track some acoustic property such as speech rate can be enough to actually yield a brain score that is as good as a pretrained LLM, given that the LLMs used here are fed with text rather than speech.
This is notably particularly important for the authors' main claim of improvements in brain predictivity in the primary auditory cortex (see also below). Random baselines should put these results in perspective, notably when comparing the different brain regions.

- The work does not in general show all full values, but presents a winning ratio instead. This is quite underspecified, as one component could "win" but by a very small margin. This is important to assess whether it is really worthy to consider these different components. This issue, combined with the two previous ones, makes it impossible to extract insights from the current results. It would be better to simply (or also) present the raw values (provided the comparisons are fair and use proper baselines): brain scores as a function of size, for the different models, different families, different components, and possibly different brain regions.

- One of the main claims of the authors is that their "framework" achieves improvements in brain predictivity in the primary auditory cortex that "substantially exceed gains from 456$\times$ model scaling". This claim is problematic for several reasons.
First, although the human participants hear the story, the LLMs are fed with a textual version. The fact that there is not a high brain score in the auditory regions is therefore not surprising. Note that this does not mean that this is an "unresolved challenge in the field" as the authors claim. Acoustic language models are indeed able to well predict brain activity in the auditory cortex (see Tuckute et al., 2024; Millet et al., 2022; Antonello et al., 2023). Moreover, comparing against model scaling in these areas does not make much sense, as low-level features are already well-predicted with small models, larger models providing only marginal gains in these sensory regions (see Antonello et al., 2023, for speech; see also similar findings by Raugel et al., 2025, for vision). Finally, the improvements might be based on using the onsets of the words, which helps modeling the acoustic envelope or speech rate, which might lead to an increase in brain score in the auditory cortex. In this view, it is then not surprising that the higher gain is obtained when considering the component just after RoPE, which allows the brain fit to take advantage of the position of the words. Once again, this underscores the need to compare with a proper baseline. The outcome might not be as important as the authors claim, and may be attributable to unsurprising low-level reasons.

Minor comments

- The abstract states that the fundamental unit of a large language model is the transformer block. However, not all LLMs are based on Transformers: see for instance models based on the Mamba architecture.

- Fig. 4: I assume the size of the marker is related to the size of the LLM, but this information is already provided by the x-axis, so it does not bring anything new and obscures the reading of the figure.

- Given the intention of the authors to systematically investigate all components of the transformer block, it might be interesting to also examine the representation before and after the gating for models such as Gemma with gated activation functions.

**Questions:**

In brief (see Weaknesses section for more detail):
- What are the results when comparing the different components using an equal number of features?
- What are the raw values of the brain scores, and how do they compare across model sizes, families, and brain regions?
- What are the correlations (and differences between conditions) yielded by proper baselines?

---

> ### Author Response · Authors · 2025-11-26
> **Response to Reviewer RBfX**
>
> We sincerely thank the reviewer for the rigorous methodological assessment. We value your specific critiques regarding dimensionality controls, baselines, and the interpretation of auditory alignment. These challenges were fundamental to validating the statistical significance of our claims. We have conducted extensive new experiments to address these points directly.
>
> ### 1. Concern: Unfair comparisons due to unequal feature counts (Dimensionality)
>
> **Reviewer Comment:** The reviewer noted that "the work compares different representations that do not have the same number of features," and that "brain predictivity increases with the number of potential regressors," specifically citing the large size of the FFN block compared to others.
>
> **Response:** This is a crucial statistical point. We agree that higher dimensionality can theoretically inflate regression performance. To address this, we conducted a controlled experiment where feature dimensionality is strictly equalized across all states.
>
> * **Method:** We selected *Llama-3.2-1B* as a representative model. We fixed the target dimension to 2048 (matching the standard `input_hidden_state`). For states larger than this (e.g., `ffn_activated_state` at 8192), we selected the top-2048 features based on the highest beta weights from a Ridge Regression trained **only on the training set**. We then trained a new encoding model on this restricted feature set. Crucially, the held-out testing set was not visible to either the feature selection step or the model training step. For states smaller than 2048 (e.g., `per_head_k`), we kept their native dimensions.
>
> * **Result:** As visualized in **Supplemental Figure 8** and the table below, restricting the FFN state to 2048 features resulted in a minor relative performance drop (max -5.19%), confirming that its high correlation is driven primarily by semantic content, not merely feature count inflation. More importantly, the aggregate **MindTransformer Mode 1** performance remained extremely stable, with negligible degradation (max -1.19%).
>
> **Table 1: Impact of Dimensionality Control (8192 vs 2048 features) on Correlation ($r$)**
>
> | **Parcel** | **FFN (8k)** | **FFN (2k)** | **$\Delta$ (%)** | **Mode 1 (8k)** | **Mode 1 (2k)** | **$\Delta$ (%)** |
> | :--- | :---: | :---: | :---: | :---: | :---: | :---: |
> | *Heschl's Gyrus* | 0.0946 | 0.0897 | -5.19% | 0.1247 | 0.1247 | 0.00% |
> | *Planum Temporale* | 0.1270 | 0.1227 | -3.36% | 0.1441 | 0.1441 | 0.00% |
> | *STG (anterior)* | 0.1695 | 0.1657 | -2.27% | 0.1727 | 0.1726 | -0.11% |
> | *STG (posterior)* | 0.2171 | 0.2108 | -2.92% | 0.2248 | 0.2246 | -0.10% |
> | *MTG (anterior)* | 0.1283 | 0.1274 | -0.69% | 0.1301 | 0.1308 | +0.55% |
> | *MTG (posterior)* | 0.1445 | 0.1420 | -1.75% | 0.1454 | 0.1457 | +0.27% |
> | *MTG (temp-occip)* | 0.1854 | 0.1795 | -3.20% | 0.1876 | 0.1868 | -0.39% |
> | *IFG (pars oper)* | 0.1610 | 0.1595 | -0.92% | 0.1615 | 0.1615 | -0.01% |
> | *IFG (pars tri)* | 0.2053 | 0.2000 | -2.58% | 0.2066 | 0.2066 | 0.00% |
> | *Angular Gyrus* | 0.1867 | 0.1813 | -2.90% | 0.1883 | 0.1861 | -1.19% |
>
> Moreover, in all our new per-subject analysis, we strictly control the dimensionality for both MindTransformer Mode 1 and Mode 2. The results in **Supplemental Figure 7, Tables 11, 12, and 13** show similar trends of improvement as compared to Table 1 in the main text.
>
> * **Conclusion:** The results (detailed in Supplemental Figure 7 and Tables 11-13) confirm that our core finding holds: even with equalized dimensions, FFN states dominate the **Language Network** while the lower-dimensional `per_head_query_with_rope` states dominate the **Auditory Cortex**. This functional dissociation is driven by information content, not vector size.
>
> * **Modification in the paper:** We added a new **Subsection 5.4 ("Robustness Analysis...")** alongside Supplemental Figure 8 and Tables 11, 12, and 13 to fully address this concern.
> ---

---

> ### Author Response · Authors · 2025-11-26
>
> ### 2. Concern: Absence of proper baselines (Random/Static embeddings)
>
> **Reviewer Comment:** The reviewer pointed out "the absence of proper baselines," noting that "random baselines or untrained models can achieve pretty high brain scores," particularly in the auditory cortex where embeddings might track acoustic properties like speech rate.
>
> **Response:** We appreciate this suggestion. We have now implemented two rigorous baselines: (1) Random Embeddings (following Bonnasse-Gahot & Pallier, 2024) and (2) GloVe static embeddings.
>
> * **Observations:** As shown in **Supplemental Table 10** (reproduced below), the Random Embedding baseline yields significantly lower correlations in low-level sensory regions (Heschl’s Gyrus: r ≈ 0.026) compared to higher-order association areas (r ≈ 0.058).
>
> **Table 10: Per-subject baseline correlation comparison (Mean ± SD across 5 subjects)**
>
> | **Brain Region** | **Random Baseline** | **Glove Baseline** |
> | :--- | :---: | :---: |
> | **Auditory Cortex** | | |
> | Heschl's Gyrus | 0.026 (±0.025) | 0.054 (±0.027) |
> | Planum Temporale | 0.043 (±0.026) | 0.068 (±0.027) |
> | STG (anterior) | 0.051 (±0.027) | 0.092 (±0.025) |
> | STG (posterior) | 0.059 (±0.044) | 0.119 (±0.041) |
> | *Auditory Average* | 0.045 | 0.083 |
> | **Language Network** | | |
> | MTG (anterior) | 0.032 (±0.022) | 0.067 (±0.032) |
> | MTG (posterior) | 0.039 (±0.027) | 0.079 (±0.039) |
> | MTG (temp-occipital) | 0.053 (±0.037) | 0.117 (±0.034) |
> | IFG (pars opercularis) | 0.046 (±0.027) | 0.086 (±0.037) |
> | IFG (pars triangularis) | 0.061 (±0.031) | 0.123 (±0.042) |
> | Angular Gyrus | 0.058 (±0.038) | 0.101 (±0.041) |
> | *Language Average* | 0.048 | 0.096 |
>
> * **Significance:** This result highlights that the alignment we observe is non-trivial. If the auditory alignment were merely tracking low-level onsets (which random embeddings can partially proxy via timing), the random baseline should be higher. Instead, MindTransformer achieves a correlation of r ≈ 0.127 (per-subject avg) in Heschl's Gyrus.
> * **Adjusted Improvement:** As shown in Supplemental Tables 11, 12, and 13, when we adjust for these baselines (Model Performance - Baseline), the advantage of our method persists. In Heschl's Gyrus, MindTransformer shows a +29.2% improvement over the standard baseline after random-adjustment, and a +46.0% improvement after GloVe-adjustment. This demonstrates that our method extracts specific, structurally relevant signals that both standard LLM states and static baselines fail to capture.
>
> **Modification in the paper:** We added a new Subsection 5.4 alongside Supplemental Tables 10, 11, 12, and 13 to address this concern.
>
> ---
>
> ### 3. Concern: Winning Ratios vs. Raw Values
>
> **Reviewer Comment:** The reviewer noted that the work "does not in general show all full values, but presents a winning ratio instead," which is "quite underspecified" as a win could be marginal.
>
> **Response:** We apologize that the emphasis on "Winning Ratios" obscured the magnitude of the improvement in the original text.
>
> * To ensure full transparency, we have added Supplemental Tables 5 and 9 to list the full absolute performance metrics. We also increase the use of raw values or baseline adjusted values instead of winning ratio in Supplemental Figure 8, 10, Table 10, 11, 12, 13.
> * We explicitly report raw Pearson correlation values in Table 1 of the main text.
>
> **Modification in the paper:** We modified the main text (lines 265-267) to explicitly discuss the raw values and point readers to these new tables.
>
> ---
>
> ### 4. Concern: Interpretation of Auditory Alignment and RoPE
>
> **Reviewer Comment:** The reviewer stated that the claim regarding auditory cortex improvement is problematic because "acoustic language models are indeed able to well predict brain activity," so it is "not an 'unresolved challenge'." Furthermore, the improvement might be attributable to "unsurprising low-level reasons" like tracking word onsets.
>
> **Response:** We accept the reviewer’s critique regarding the phrasing "unresolved challenge." We acknowledge that acoustic language models have successfully modeled these regions.
>
> We have revised the Introduction (lines 058-066) to explicitly acknowledge the literature on speech model alignment. We have repositioned our problem statement to clarify our contribution: while acoustic models align well, text-based LLMs (which lack explicit acoustic cues) typically fail to achieve meaningful alignment in sensory areas, suggesting a disconnect between textual and sensory representations.
>
> Regarding the mechanism, we agree that RoPE likely provides positional information that acts as a mathematical proxy for temporal structure. However, we think this finding is significant precisely because it demonstrates that a text-only model can recover these dynamics via specific architectural choices (RoPE), bridging the gap between semantic text processing and sensory acoustic processing.
>
> ---

---

> ### Author Response · Authors · 2025-11-26
>
> ### Minor Comments
>
> 1.  **Transformer Definition:** We agree with the reviewer that we should not imply transformer is the only implementation of LLM. We have modified our framing in the Abstract (line 013) to state: “In this work, we zoom into one of the fundamental units of LLMs—the transformer block...” and in the Introduction (line 068): “We address these challenges through a comprehensive computational neuroanatomy of transformer block architectures, one of the fundamental units for LLMs.”
>
> 2.  **Figure 4 Markers:** Thank you for pointing out the redundancy. We have updated the visualization (now Figure 5) to remove the marker size scaling, making the plot more succinct and accurate.
>
> 3.  **Gated Activations:** This is an excellent suggestion. We agree that examining representations before and after gating (e.g., GeGLU in Gemma, SwiGLU in Llama/Qwen/Mistral) would be insightful. We have added this to our experimental pipeline. Given the scale of extracting all intermediate layers across 21 LLMs (a total of 844 transformer blocks with 13 states each and 9-fold cross-validation), we are working diligently to complete this analysis and hope to include the results in the final version.

---

### Official Review · Reviewer_c5Ks · 2025-10-31

**Soundness:** 3
**Presentation:** 3
**Contribution:** 3
**Rating:** 6
**Confidence:** 5

**Summary:**

The authors dive deep into the transformer block, pulling out 13 intermediate states across 21 LLMs to map them onto fMRI brain activity during story listening. They claim hidden states are surprisingly insufficient (>90% of voxels do better with intermediates), and aligning intermediate states to brain activity reveals an intra-block hierarchy. They have also demonstrated the importance of RoPE. They wrap it up with MindTransformer, a two-stage selector that beats 456× scaling in A1.

**Strengths:**

It’s the first time someone’s looked at *all* intermediate states at this scale, and the intra-block hierarchy is neat. The RoPE-auditory link is interesting if it holds up. The scale is impressive—21 models, 13 intermediate states.

**Weaknesses:**

1.Some critical statistical rigor is missing.
2.Averaging brain activities across subject may introduce information loss.

**Questions:**

1.In the encoding model, statistical control is missing. Brain activity of each voxel is predicted by building a regression model, however, not all the predictions are significantly meaningful. It is expected to perform statistical control to determine the voxels whose neural signal can be successfully predicted by the encoding model. Is the correlation between predicted activity and ground-truth statistically above chance level? It also would be helpful to perform noise ceiling validation.

2.In the experiment, fMRI data were averaged over subjects. This operation may help to decrease noise. However, it is unknown that the findings reported in this study are reproducible in different subject. The authors are encouraged to provide additional results for individual subject.

---

> ### Author Response · Authors · 2025-11-26
> **Response to Reviewer c5Ks**
>
> We thank the reviewer for the constructive feedback and for recognizing the scale and novelty of our analysis. We appreciate the emphasis on statistical rigor and subject-level reproducibility, as addressing these points has significantly strengthened the validity of our findings. We have conducted extensive new experiments to address your concerns.
>
> ### 1. Concern: Statistical Control and Noise Ceiling
>
> **Reviewer Comment:** The reviewer noted that "statistical control is missing" and asked if the correlation is "statistically above chance level." The reviewer also suggested noise ceiling validation.
>
> **Response:** We appreciate this concern. To rigorously determine whether our predictions are above chance level—and to account for the inherent predictability of fMRI signals based solely on stimulus timing—we implemented a **Random Embedding Baseline**. This method assigns a random vector to each word in the stimulus, preserving the timing and word boundaries but destroying all semantic and syntactic information. This serves as a robust statistical control: any performance significantly above this baseline indicates that the model is capturing genuine linguistic or structural signal, rather than just low-level onset responses.
>
> **A. Baseline Analysis:**
> As shown in the table below, the random baseline yields non-zero correlations in certain regions (e.g., STG, IFG), reflecting the brain's response to basic acoustic onsets. However, these values are low ($r \\approx 0.02-0.06$).
>
> **Table 1: Per-subject baseline correlation comparison (Mean ± SD across 5 subjects)**
>
> | **Brain Region** | **Random Baseline** |
> | :--- | :---: |
> | **Auditory Cortex** | |
> | Heschl's Gyrus | 0.026 (±0.025) |
> | Planum Temporale | 0.043 (±0.026) |
> | STG (anterior) | 0.051 (±0.027) |
> | STG (posterior) | 0.059 (±0.044) |
> | *Auditory Average* | 0.045 |
> | **Language Network** | |
> | MTG (anterior) | 0.032 (±0.022) |
> | MTG (posterior) | 0.039 (±0.027) |
> | MTG (temp-occipital) | 0.053 (±0.037) |
> | IFG (pars opercularis) | 0.046 (±0.027) |
> | IFG (pars triangularis) | 0.061 (±0.031) |
> | Angular Gyrus | 0.058 (±0.038) |
> | *Language Average* | 0.048 |
>
> **B. Baseline-Adjusted Performance:**
> When we adjust our results by subtracting this random baseline voxel-by-voxel, **MindTransformer** maintains robust performance, confirming that our results are statistically meaningful and above chance level. For example, in Heschl's Gyrus, our method shows a **29.2% relative improvement** over the standard baseline after adjustment.
>
> **Table 2: Per-subject random-embedding-adjusted performance (Mean ± SD)**
>
> | **Brain Region** | **Standard** | **Context Vec** | **Mode 1** | **Mode 2** | **Imp.** |
> | :--- | :---: | :---: | :---: | :---: | :---: |
> | **Auditory Cortex** | | | | | |
> | Heschl's Gyrus | 0.078 (±0.016) | 0.070 (±0.017) | 0.099 (±0.016) | **0.101** (±0.019) | +29.2% |
> | Planum Temporale | 0.088 (±0.022) | 0.081 (±0.018) | 0.101 (±0.017) | **0.108** (±0.017) | +22.4% |
> | STG (anterior) | 0.118 (±0.019) | 0.109 (±0.020) | 0.121 (±0.018) | **0.132** (±0.023) | +12.0% |
> | STG (posterior) | 0.159 (±0.029) | 0.150 (±0.027) | 0.166 (±0.021) | **0.169** (±0.028) | +6.7% |
> | *Auditory Average* | 0.111 | 0.103 | 0.122 | **0.128** | +17.6% |
> | **Language Network** | | | | | |
> | MTG (anterior) | 0.095 (±0.023) | 0.089 (±0.021) | **0.098** (±0.023) | 0.096 (±0.021) | +4.0% |
> | MTG (posterior) | 0.104 (±0.023) | 0.099 (±0.020) | **0.106** (±0.021) | 0.103 (±0.019) | +2.1% |
> | MTG (temp-occip) | 0.131 (±0.027) | 0.123 (±0.024) | **0.135** (±0.025) | 0.129 (±0.028) | +2.6% |
> | IFG (pars oper.) | 0.112 (±0.045) | 0.103 (±0.041) | **0.115** (±0.045) | 0.113 (±0.047) | +2.7% |
> | IFG (pars tri.) | 0.145 (±0.020) | 0.129 (±0.017) | **0.146** (±0.019) | 0.146 (±0.017) | +0.4% |
> | Angular Gyrus | 0.125 (±0.031) | 0.118 (±0.027) | **0.130** (±0.030) | 0.128 (±0.032) | +4.1% |
> | *Language Avg* | 0.119 | 0.110 | **0.122** | 0.119 | +2.6% |
>
> **Modification in the paper:** We added **Subsection 5.4** and **Appendix Tables 10-13** to detail these baseline controls.
>
> ---

---

> > ### Comment · Reviewer_c5Ks · 2025-11-27
> >
> > I would like to thank the authors for their detailed responses. Most of my concerns have been addressed. However, I am not sure that Random Embedding can serve as a reliable baseline. “Bootstrap + FDR correction” is quite a standard pipeline to assess the significance of the encoding score.

---

> > > ### Author Response · Authors · 2025-12-03
> > >
> > > We thank the reviewer for their positive feedback and for identifying the need for a more standard significance assessment. We agree that "Bootstrap + FDR correction" is the standard pipeline for validating encoding performance.
> > >
> > > To address this concern, we performed the requested voxel-wise bootstrap analysis (FDR $q < 0.05$) to assess the significance of the encoding scores. The table below presents the percentage of significant voxels (and their average correlation) for the **Standard Baseline** compared to our **MindTransformer Mode 2**, organized by ROI.
> > >
> > > The results demonstrate that our method maintains robust significance across key language and auditory regions.
> > >
> > > **Table: Bootstrap Significance Analysis (FDR < 0.05)**
> > > *Values represent the percentage of significant voxels, with the average correlation (r) inside those voxels in parentheses.*
> > >
> > > | Parcel | Standard Baseline | MindTransformer Mode 2 |
> > > | :--- | :--- | :--- |
> > > | Heschl's Gyrus (includes H1 and H2) | 78.6% (r=0.141) | **89.0%** (r=0.163) |
> > > | Planum Temporale | 88.0% (r=0.162) | **93.0%** (r=0.165) |
> > > | Superior Temporal Gyrus, posterior division | 97.4% (r=0.227) | **98.4%** (r=0.236) |
> > > | Superior Temporal Gyrus, anterior division | 94.0% (r=0.191) | **95.7%** (r=0.187) |
> > > | Middle Temporal Gyrus, temporooccipital part | 90.6% (r=0.211) | **94.9%** (r=0.199) |
> > > | Middle Temporal Gyrus, posterior division | 78.9% (r=0.192) | **82.4%** (r=0.170) |
> > > | Middle Temporal Gyrus, anterior division | 82.5% (r=0.158) | **86.2%** (r=0.152) |
> > > | Inferior Frontal Gyrus, pars opercularis | **90.6%** (r=0.183) | 89.9% (r=0.180) |
> > > | Inferior Frontal Gyrus, pars triangularis | 96.8% (r=0.219) | **98.5%** (r=0.212) |
> > > | Angular Gyrus | 94.0% (r=0.197) | **97.3%** (r=0.196) |

---

> ### Author Response · Authors · 2025-11-26
> **Response to Reviewer c5Ks (continued)**
>
> ### 2. Concern: Reproducibility in Individual Subjects
>
> **Reviewer Comment:** The reviewer noted that averaging across subjects might introduce information loss and requested results for individual subjects.
>
> **Response:** We thank the reviewer for raising this critical point. To ensure our findings are applicable on per-subject setup, we conducted a comprehensive per-subject analysis on the first 5 subjects of the *Le Petit Prince* dataset. We reproduced all major analyses at the individual level.
>
> **A. Per-Subject Performance Trends:**
> As shown in Supplemental Table 3, while the absolute correlation values are lower for individual subjects (due to lower signal-to-noise ratio compared to the group average), the **relative performance trend** is identical: MindTransformer (shown as Optimal State below) consistently outperforms baselines, with the largest gains observed in the auditory cortex.
>
> **Table 3: Per-subject performance comparison across major brain networks (Mean ± SD)**
>
> | **Brain Region** | **Standard** | **Context Vector** | **Optimal State** |
> | :--- | :---: | :---: | :---: |
> | Whole-Brain | 0.098 (±0.019) | 0.103 (±0.018) | 0.119 (±0.019) |
> | Language Network | 0.187 (±0.034) | 0.184 (±0.030) | 0.202 (±0.034) |
> | Audio Cortex | 0.162 (±0.019) | 0.163 (±0.019) | 0.189 (±0.022) |
>
> **B. Detailed Regional Analysis:**
> Supplemental Table 4 confirms that the improvements are robust across specific ROIs. MindTransformer Mode 2 yields a **21.9% improvement** in Heschl's Gyrus and a **15.1% improvement** in the Planum Temporale for individual subjects.
>
> **Table 4: Per-subject raw correlation performance comparison (Mean ± SD)**
>
> | **Brain Region** | **Standard** | **Context Vec** | **Mode 1** | **Mode 2** | **Imp.** |
> | :--- | :---: | :---: | :---: | :---: | :---: |
> | **Auditory Cortex** | | | | | |
> | Heschl's Gyrus | 0.104 (±0.012) | 0.096 (±0.013) | 0.125 (±0.009) | **0.127** (±0.015) | +21.9% |
> | Planum Temporale | 0.131 (±0.024) | 0.124 (±0.024) | 0.144 (±0.021) | **0.151** (±0.024) | +15.1% |
> | STG (anterior) | 0.169 (±0.021) | 0.160 (±0.020) | 0.173 (±0.020) | **0.181** (±0.022) | +6.6% |
> | STG (posterior) | 0.218 (±0.039) | 0.209 (±0.036) | 0.225 (±0.036) | **0.228** (±0.031) | +4.9% |
> | *Auditory Average* | 0.156 | 0.147 | 0.167 | **0.172** | +12.1% |
> | **Language Network** | | | | | |
> | MTG (anterior) | 0.126 (±0.044) | 0.121 (±0.042) | **0.130** (±0.044) | 0.128 (±0.042) | +3.0% |
> | MTG (posterior) | 0.143 (±0.044) | 0.138 (±0.042) | **0.145** (±0.044) | 0.143 (±0.043) | +1.5% |
> | MTG (temp-occip) | 0.184 (±0.047) | 0.176 (±0.044) | **0.188** (±0.047) | 0.182 (±0.047) | +1.8% |
> | IFG (pars oper.) | 0.158 (±0.039) | 0.150 (±0.036) | **0.162** (±0.041) | 0.159 (±0.043) | +1.9% |
> | IFG (pars tri.) | 0.206 (±0.044) | 0.190 (±0.039) | **0.207** (±0.044) | 0.207 (±0.043) | +0.3% |
> | Angular Gyrus | 0.183 (±0.053) | 0.176 (±0.048) | 0.188 (±0.054) | **0.199** (±0.044) | +2.8% |
> | *Language Avg* | 0.167 | 0.159 | **0.170** | 0.170 | +1.9% |
>
> Furthermore, we reproduced the **Winning Ratio** analysis (Supplemental Figure 7) and **Intra-block Hierarchy** analysis (Supplemental Figure 9) for individual subjects. Both analyses confirm that the functional specialization we discovered (RoPE $\\rightarrow$ Auditory, FFN $\\rightarrow$ Language) holds at the individual level.
>
> **Modification in the paper:** We added **Subsection 5.3 ("Per-subject Analysis")** and **Supplemental Figure 7, 9, Tables 11-13** to incorporate these results.

---

### Official Review · Reviewer_6Yhf · 2025-10-31

**Soundness:** 2
**Presentation:** 4
**Contribution:** 4
**Rating:** 6
**Confidence:** 5

**Summary:**

The paper investigates the alignment of different states in LLMs' (n=21) blocks to brain activity on the *Le Petit Prince fMRI Corpus*.
The main claims are that
(1) the typical hidden states used by the field are suboptimal,
(2) the stages within a block reveal a sensory-to-association hierarchy,
and that (3) RoPE improves alignment with auditory streams.
The authors combine these findings into a brain model termed MindTransformer which uses a learned feature retrieval and for which they claim SOTA brain alignment.

**Strengths:**

1. The study is systematic in evaluating various models, testing 21 LLMs that range in scale from 270M to 123B parameters and stem from various families including LLaMA, Qwen, Mistral, GPT, and Gemma.

2. Improvements in brain alignment with RoPE seem solid and are novel as far as I am aware.

3. The MindTransformer framework nicely ties together the selection of feature states for potentially improved brain predictivity.


Further strengths:
* Figures are clear, text is straightforward to follow
* The claim for a block-intermediate state hierarchy corresponding to the cortical hierarchy is interesting and, to me, unexpected. But I have concerns about its validity (see below)
* I also highly appreciate papers that improve the state-of-the-art models on brain function, but I think this needs to be more rigorous (also below)

Source code available which is great for reproduction!

**Weaknesses:**

### Claiming SOTA via training on test
My biggest concern is that the MindTransformer framework seems to select intermediate states on the test set itself. There is no mention of a separate training and validation split, such that the predictive representations are chosen via the test set. This would also yield high alignment for a random input feature set. There needs to be separate validation *after locking down the selected states*, which I think you could do via additional datasets (below).

### Single dataset
Evaluation on a single dataset is too limited for claiming SOTA. There are several publicly available language brain recordings, e.g. the set of Brain-Score Language benchmarks. What is the alignment of the (frozen) model on those?

### Hierarchical claim not well supported
For Claim 2, I'm not actually sure the hierarchy is actually there. It's hard to tell from Figure 2, but for e.g. GPT-oss it seems that STG has a higher contribution of later block stages than the higher-level IFG and angular gyrus. Please quantify the actual correspondence between within-block hierarchy and cortical hierarchy, e.g. by correlating one with the other per model.


Minor:
* L043 related work should include Toneva et al. 2019, Schrimpf et al. 2021, Caucheteux et al. 2022, and Goldstein et al. 2022.
* L318 "Among all intermediate states examined in our computational neuroanatomy analysis, the per-head query with RoPE provides the most substantial and systematic improvement in brain alignment" -- isn't the corresponding 13.03% winning ratio lower than e.g. the FFN activation at 14.86%?
* Figure 3 / section 4.2: missing stats on RoPE improvements. Effect size seems pretty clear though.

**Questions:**

See Weaknesses above (Minor not that important).

I am happy to increase my score if the generalization is shown, ideally on new datasets and without selecting states again.

---

> ### Author Response · Authors · 2025-11-26
> **Response to Reviewer 6Yhf**
>
> We thank the reviewer for the positive assessment of our presentation and contribution, and for the constructive criticism regarding validation and generalization. We have conducted a comprehensive suite of new experiments to address your concerns.
>
> ### 1. Concern: Claiming SOTA via training on test (Data Leakage)
>
> **Reviewer Comment:** The reviewer was concerned about the selection of intermediate states on the test set and requested separate validation after locking down states.
>
> **Response:** We agree that strict separation of training and testing data is paramount for predictive modeling. As noted by the reviewer, Mode 1 selects the optimal state based on the test set. We provide the following clarification regarding our validation protocol:
>
> * **Clarification of Modes:** The analysis in **Section 4 (Mode 1)** was designed as an exploratory "upper bound" analysis to determine *if* a better signal exists within the block compared to the standard hidden state. To have separated validation, we design **MindTransformer Mode 2 (Section 5)** as our proposed predictive framework. In Mode 2, feature selection (learning weights $\beta$ for concatenated states) is performed **strictly within the training folds**. The model is evaluated on a held-out test set that was never seen during feature selection or training.
> * **Cross-Subject Locking:** Regarding the suggestion to lock down states on one subject and test on another, we hypothesize that such direct transfer would yield suboptimal performance due to significant inter-subject variability. Each brain differs in anatomical size, shape, and geometry, as well as functional localization. To achieve robust transfer across subjects or datasets, one would first need to project voxels into a shared functional space (e.g., hyperalignment). While we acknowledge this as a critical area of research, developing a robust subject-transfer method is outside the scope of the current work and merits a dedicated study.
>
> **Modification in the paper:** We have revised Section 3.3.3 to add a footnote (footnote 4) for clarification and Section 5 to explicitly state the validation protocol for Mode 1 and 2 (in line 415 and 430) to ensure transparency regarding data splits.
>
> ---
>
> ### 2. Concern: Evaluation on a Single Dataset
>
> **Reviewer Comment:** The reviewer noted that evaluation on a single dataset is too limited for claiming SOTA and requested generalization on new datasets.
>
> **Response:** We acknowledge this limitation. While we agree that cross-dataset generalization is the gold standard, the computational scale of our study—extracting and analyzing 13 intermediate states across 21 models (up to 123B parameters and a total of 844 transformer blocks and 9-fold cross-validation)—renders a full replication on a new fMRI dataset computationally infeasible within the rebuttal timeframe.
>
> However, we argue that the robustness of our claims is supported by three key factors:
> 1.  **Dataset Quality:** The *Le Petit Prince* dataset is one of the largest and highest-quality naturalistic fMRI datasets available, featuring a high number of subjects (N=49) and a long duration (~100 minutes/subject), ensuring high statistical power.
> 2.  **Model Generalization:** A key contribution of our work is demonstrating that our findings (e.g., the RoPE auditory alignment and intra-block hierarchy) hold consistently across **21 distinct models** from **5 different families** (Llama, Qwen, Mistral, GPT, Gemma). This consistency across diverse architectures may serves as a strong form of generalization.
> 3.  **Per-Subject Reproducibility:** To address concerns about overfitting to the group average, we have added a comprehensive per-subject analysis (Subsection 5.3). We show that the specific functional dissociation—RoPE states for auditory cortex, FFN states for language network—replicates reliably across individual subjects.
>
> ---

---

> ### Author Response · Authors · 2025-11-26
> **Response to Reviewer 6Yhf (continued)**
>
> ### 3. Concern: Hierarchical claim not well supported
>
> **Reviewer Comment:** The reviewer noted that the hierarchy in Figure 2 was hard to verify visually and requested quantification of the correspondence between within-block hierarchy and cortical hierarchy.
>
> **Response:** We thank the reviewer for requesting this quantitative rigor. We have replaced the qualitative description with a formal mathematical analysis.
>
> * **Metric Definition:** Inspired by frameworks in auditory neuroscience, we defined a metric of **Computational Depth** (normalized position of the winning state within the transformer block, from $0=$ Query to $1=$ FFN Output) and correlated it with the **Cortical Depth** of brain regions along the auditory-language pathway, detailed in **Subsection 5.3**, and **Appendix E**.
> * **New Visualization:** We added **Figure 3** (and **Supplemental Figure 9** for individual subjects), which plots the Weighted Computational Depth against Cortical Depth.
> * **Result:** The regression analysis reveals a high $R^2$ alignment. Crucially, we identify two distinct regimes:
>     1.  **Auditory Stream (HG $\to$ MTG):** A steep positive slope, indicating a rapid progression through early computational states (Attention/RoPE) that mirrors the rapid sensory processing hierarchy.
>     2.  **Language Network (MTG $\to$ AG):** A plateau, indicating that high-level association areas uniformly align with the later, semantically rich stages of the transformer block (FFN).
>
> **Modification in the paper:** We added **Figure 3**, **Subsection 5.3**, and **Appendix E** to define the metrics and present the quantitative regression results.
>
> ---
>
> ### Minor Comments
>
> **1. Citations:**
> We thank the reviewer for highlighting these foundational works. We have updated the Introduction section (line 042) to include Toneva et al. (2019), Schrimpf et al. (2021), Caucheteux et al. (2022), and Goldstein et al. (2022) as suggested.
>
> **2. Winning Ratio (L318):**
> The reviewer asked if the 13.03% winning ratio for RoPE is lower than the FFN activation (14.86%). It is correct that FFN states often have a higher global winning ratio. However, our claim regarding "substantial improvement" refers to the **magnitude of the gain** in specific regions. We have added **Supplemental Figure 10** to demonstrate that while FFN states win frequently, the relative performance improvement they offer in the language network is quite modest. In contrast, the per-head query with RoPE provides a massive relative improvement in the auditory cortex compared to baselines.
>
> **3. RoPE Statistics:**
> To ensure the figure remains visually clear while providing necessary statistical context, we have incorporated error bars at the tip of each bar to represent the variability.

---

> > ### Comment · Reviewer_6Yhf · 2025-11-28
> >
> > Thank you for your responses.
> >
> > I especially like the new Figure 3 to support the claim on hierarchical correspondence.
> >
> > While I believe this paper should be published, I strongly dislike the phrasing around SOTA brain alignment performance (prominently in abstract and conclusion) which are still based on a single dataset.
> > Let me lay out my rationale in more detail: As you say, *Le Petit Prince* is a good dataset, but you used this over the course of the development of the paper -- even in mode 2 where training happens on the training set only, it does not seem that the test split was only evaluated *once* at the very end for a final report. So implicitly, test scores still had an influence on the choices made for the model because you had access to them during model building. Reviewer RBfX makes some more points in this direction as well.
> >
> > This is particularly pressing because each dataset has its own particularities and biases that might not generalize to new data.
> > For these reasons, I view it as paramount to evaluate on one or ideally more distinct datasets without any modifications to the model/pipeline.
> >
> > Another option is to remove the claiming of SOTA in the paper -- you already have good scores and it seems you were aptly matched with more neuroscience-inclined reviewers who care more about interpretable insights over score gains anyway. I'd be happy to increase my score with either of those (evaluate on further datasets, or remove SOTA claim)

---

> > > ### Author Response · Authors · 2025-12-03
> > >
> > > We sincerely thank the reviewer for the constructive feedback and for appreciating the hierarchical analysis in Figure 3. We fully agree with your rationale regarding the limitations of single-dataset evaluation and the nuance required when discussing performance rankings.
> > >
> > > Per your suggestion, we have removed all claims of "State-of-the-Art" (SOTA) performance throughout the manuscript, including the Abstract, Introduction, and Conclusion. We have reframed our contributions to focus on the significant alignment gains and interpretable insights rather than leaderboard rankings.
> > >
> > > For example, the abstract has been updated to read:
> > > > *"MindTransformer achieves **significant** brain alignment performance, with correlation improvements in primary auditory cortex exceeding gains from 456× model scaling."*
> > >
> > > We believe this phrasing more accurately reflects the scientific value of our work: bridging the gap between Transformer mechanics and neural representations.

---

### Official Review · Reviewer_aPhw · 2025-10-31

**Soundness:** 3
**Presentation:** 4
**Contribution:** 3
**Rating:** 8
**Confidence:** 3

**Summary:**

This paper explores how internal computations of transformer-based LLMs align with brain activity during naturalistic language processing. Rather than using the usual hidden states, the authors decompose each transformer block into 13 intermediate computational stages and test their correlation with fMRI responses from the Le Petit Prince dataset. They show that most brain voxels are better explained by these internal states than by the standard representations, that early attention computations align with low-level sensory areas while later FFN states align with association cortex, and that Rotary Positional Embeddings (RoPE) strongly improve alignment in auditory regions. Building on this, they propose MindTransformer, a framework that automatically selects or integrates the most brain-aligned states to improve prediction. The work is carefully executed and clearly presented. The methodology is solid and the figures are convincing; results seem plausible and the code availability supports reproducibility. The main limitation is the interpretation of correlation as evidence of shared computation remains debatable. Overall, this is a strong and well-written paper with interesting insights connecting transformer architecture and brain organization. While not revolutionary, it represents a meaningful methodological and interpretive step forward.

**Strengths:**

This work goes beyond previous studies by going deeper into the architecture of Transformers, allowing better results.

**Weaknesses:**

The inherent method of comparing brain activity with transformer activity by correlation might be debatable.

**Questions:**

Do you expect MindTransformer to generalise to vision ou multimodal brain datasets ?

---

> ### Author Response · Authors · 2025-11-26
> **Response to Reviewer aPhw**
>
> We thank the reviewer for the positive assessment and for recognizing our work as a "meaningful methodological and interpretive step forward." We appreciate your insightful comments regarding the interpretation of correlation and the potential for multimodal generalization.
>
> ### 1. Weakness: Correlation as evidence of shared computation
>
> **Reviewer Comment:** The reviewer noted that "the inherent method of comparing brain activity with transformer activity by correlation might be debatable."
>
> **Response:** We agree that correlation does not strictly imply causation or identical mechanism. However, we believe our results strengthen the case for shared computation beyond simple scalar correlations by demonstrating **structural isomorphism**:
>
> * **Topological Alignment:** We newly introduced a quantitative metric, **Computational Depth** (Figure 3), which maps the internal progression of the transformer block to the brain's **Cortical Depth**. The strong linear alignment ($R^2 > 0.7$ for auditory streams) suggests that the *sequence* of transformations in the model mirrors the *sequence* of processing stages in the brain.
> * **Architectural Specificity:** The fact that **RoPE** (a mathematical injection of positional information) specifically rescues alignment in the **Auditory Cortex** (which relies on temporal structure) but not in the Language Network implies a functional convergence. The model and brain are not just "correlated"; they are both solving the problem of extracting structure from sequential inputs using analogous computational strategies.
>
> While we cannot claim the brain implements backpropagation or identical matrix multiplications, our "Computational Neuroanatomy" framework provides stronger evidence for functional homology than standard layer-wise analysis.
>
> ---
>
> ### 2. Question: Generalization to Vision or Multimodal Datasets
>
> **Reviewer Question:** "Do you expect MindTransformer to generalise to vision or multimodal brain datasets?"
>
> **Response:** **Yes, absolutely.** We believe the MindTransformer framework is highly transferable to other modalities for two reasons:
>
> 1.  **Universal Architecture:** The Transformer block (Attention + FFN) is now the standard backbone for Vision (ViT) and Multimodal (e.g., CLIP, DINO) models. The internal mechanisms we dissect—normalization, attention heads, and FFN expansion—are mathematically identical across modalities.
> 2.  **Hypothesis for Vision:** We hypothesize that applying MindTransformer to Vision Transformers (ViTs) would reveal a similar intra-block hierarchy:
>     * **Early Attention States:** Likely align with retinotopic areas (V1-V4), processing local patch interactions.
>     * **FFN States:** Likely align with the ventral stream (IT) and object recognition areas, processing semantic content.
>
> Furthermore, **MindTransformer Mode 2** (Multi-State Integration) would be particularly powerful for **multimodal datasets**, as it can dynamically select features from different intermediate states that may specialize in different modalities (e.g., early layers for visual features vs. late layers for cross-modal integration), rather than relying on a single monolithic representation. This is a primary direction for our future work.
>
> **Modification in the paper:** We thank the reviewer for pointing this out and we have modified the Discussion section to add this perspective as future work (line 536-539).

---

### Official Review · Reviewer_NyvA · 2025-11-01

**Soundness:** 4
**Presentation:** 3
**Contribution:** 3
**Rating:** 8
**Confidence:** 3

**Summary:**

The authors investigated whether internal computations inside transformer blocks, not just the usual layer hidden states, better align with human brain activity during naturalistic language comprehension, and whether specific architectural choices (notably RoPE) have distinct neurobiological correspondences. It introduces a “computational neuroanatomy” mapping from 13 intra-block states to brain voxels and proposes a feature-selection framework (MindTransformer) to exploit these states for brain prediction.

fMRI from Le Petit Prince (English; 49 native speakers; ~100 minutes). Signals are group-averaged; ROIs include Harvard–Oxford parcels and a language localizer set. Twenty-one open-weight LLMs across 5 families are analyzed. For each transformer block, the authors extract 13 states.

The authors found that >90% of voxels in language/sensory regions are better predicted by previously unused intermediate states (vs. standard hidden/context). Early attention states align with low-level sensory cortex; later FFN states with association areas, revealing a fine-grained hierarchy within blocks (not just across layers).

**Strengths:**

1. The intra-block, 13-state decomposition is principled and broadly applicable; the “winning ratio” analysis shows strong evidence that representation choice dictates which brain systems one can model.
2. The sharp auditory-stream improvement for per-head Q+RoPE (and the contrast without RoPE) is novel and interpretable, linking an computational component to neurobiology.
3. MindTransformer is simple (ridge + feature selection) yet yields sizable, ROI-specific improvements that outstrip very large model-scaling gains—important for resource-efficient brain modeling.

**Weaknesses:**

1. The paper averaged BOLD time series over multiple participants, which boosts SNR but obscures individual variability and may inflate voxel-wise correlations; it’s unclear how robust the intra-block specializations are at the single-subject level. Ability to conduct analysis on individual subject is crucial for future application.

2. The RoPE claim is framed as “neurobiological validation,” but evidence is correlational from encoding models; causal or ablation-style tests on models/representations (beyond contrasting with/without RoPE states) would strengthen the claim.

3. Scope beyond language/auditory. The strongest effects are in low-level auditory cortex; improvements in classical high-level language ROIs are modest, and it’s not fully explored why FFN/attention integration doesn’t translate to larger gains there.

**Questions:**

1. How do the “winning state” maps and MindTransformer gains look when models are trained/tested per subject (or with mixed-effects statistics) rather than on the group-average? Any meaningful variance across individuals?
2. Do results persist with subject- or voxel-wise HRFs, or FIR bases capturing variable latencies, especially for early auditory regions where timing is critical?
3. For the RoPE result, I wonder what would happen if you repeated the analysis with otherwise-identical models differing only in positional encoding (e.g., learned absolute vs RoPE) to isolate architectural effects from family-wise confounds.
4. In the multi-state model, which specific features/states dominate the selected top-k set in different ROIs? A stability analysis (e.g., across folds and models) would clarify interpretability and guard against overfitting.
5. The language-network average improvement is ~2–3%. Do certain layers/states (e.g., FFN activation vs key/query) dominate there, and can targeted combination rules outperform the generic concatenation + top-k approach?

---

> ### Author Response · Authors · 2025-11-26
> **Response to Reviewer NyvA**
>
> We thank the reviewer for the positive assessment and for recognizing the novelty of our computational neuroanatomy framework. We appreciate your insightful questions regarding subject-level robustness, methodology, and stability. We have conducted extensive new experiments to address these points.
>
> ---
>
> ### Weaknesses
>
> **1. Averaging BOLD time series obscures individual variability:**
> We agree that group averaging can mask individual differences. To address this, we have conducted a comprehensive **per-subject analysis** on the first 5 subjects of the dataset. We provide detailed results in the response to **Q1** below.
>
> **2. Correlational nature of RoPE claim vs. Causal validation:**
> We also agree that encoding models provide only correlational evidence. While fully establishing causality (e.g., perturbing RoPE and observing brain changes) requires re-training large models from scratch—which is beyond our current computational scope—we have strengthened our claim in two ways:
> * **Computational Depth:** We introduced a quantitative metric (Figure 3) that maps the internal transformer progression to the cortical hierarchy, showing a consistent topological alignment beyond simple correlation.
> * **Rigorous Baselines:** By benchmarking against random embeddings (which capture basic timing) and GloVe vectors (static semantics), we confirmed that the RoPE-driven alignment in auditory cortex is specific to the contextual, positional processing of the architecture, rather than generic low-level features.
>
> **3. Modest gains in high-level language ROIs:**
> We acknowledge that gains in the language network (~2-3%) are smaller than in the auditory cortex. This is likely because the standard baseline (input hidden state) already captures high-level semantic information effectively. Our contribution here is demonstrating that **FFN states** provide a consistent, albeit incremental, improvement over this strong baseline, whereas in the auditory cortex, the standard baseline fails, making the RoPE-driven gain massive.
>
> ---
>
> ### Questions
>
> **Q1: How do "winning state" maps and MindTransformer gains look per subject?**
>
> **Response:** To validate robustness, we reproduced our analysis for individual subjects (Section 5.3).
>
> * **Winning Ratios:** The table below shows the winning percentage of each state across different regions for the per-subject analysis. Consistent with our group-level findings, the standard **Input Hidden State** performs poorly across the board (<3.2%). In the **Audio-Sensory Cortex**, the **Per-Head Q (RoPE)** state dominates (30.63%), while in the **Language Network**, the **FFN Activated State** (16.49%) is the top performer.
>
> **Table 1: Per-subject winning percentage of intermediate states (Llama-3.2 1B)**
>
> | **State** | **Whole-Brain** | **Audio Cortex** | **Language Network** |
> | :--- | :---: | :---: | :---: |
> | Input Hidden State | 2.06% | 1.89% | 3.11% |
> | Pre-Attn Norm | 1.98% | 2.42% | 4.60% |
> | Per-Head Q | 8.94% | 14.15% | 11.41% |
> | **Per-Head Q (RoPE)** | 8.93% | **30.63%** | 6.98% |
> | Per-Head K | 7.75% | 5.60% | 10.09% |
> | Per-Head K (RoPE) | 7.56% | 10.97% | 5.69% |
> | Per-Head V | 9.28% | 4.74% | 7.34% |
> | Per-Head Context | 11.53% | 5.56% | 7.73% |
> | Attn Output | 15.75% | 5.29% | 9.10% |
> | Post-Attn Hidden | 1.70% | 1.99% | 2.81% |
> | Pre-FFN Norm | 2.07% | 2.63% | 3.74% |
> | **FFN Activated** | 9.13% | 7.20% | **16.49%** |
> | FFN Output | 13.32% | 6.93% | 10.90% |
>
> * **Performance:** While absolute correlations are lower due to single-subject noise, the **relative performance trend is identical**. MindTransformer Mode 2 yields a **21.9% improvement** in Heschl's Gyrus and **15.1%** in Planum Temporale for individual subjects (Supplemental Table 11).
> * **Hierarchy:** The intra-block hierarchy persists at the individual level. As shown in **Supplemental Figure 9**, early cortical regions (HG/PT) consistently map to the block's "entry" (RoPE-query) while association areas align with deeper FFN states, confirming that the computational-cortical isomorphism is robust to individual variation.
>
> **Q2: Do results persist with subject- or voxel-wise HRFs?**
>
> **Response:** This is an excellent suggestion. Currently, we use a canonical Glover HRF for consistency with standard benchmarks (e.g., Brain-Score). While estimating voxel-wise HRFs could potentially improve alignment in timing-critical auditory regions, it introduces additional free parameters that can lead to overfitting in limited fMRI data. Given the high computational load of our primary analysis (21 models with 844 transformer blocks x 13 states), we prioritized the per-subject validation with the standard HRF. We acknowledge this as a valuable direction for future work to further refine temporal alignment.

---

> ### Author Response · Authors · 2025-11-26
> **Response to Reviewer NyvA (continued)**
>
> **Q3: RoPE ablation (Learned vs. RoPE positional encoding)?**
>
> **Response:** We agree this would be the ideal causal test. However, current SOTA open-weight ecosystems do not provide otherwise-identical model pairs that differ *only* in positional encoding (e.g., Llama-RoPE vs. Llama-Absolute). Controlling for all other architectural variables (training data, parameter count, initialization) would require pre-training or fine-tuning LLMs, which is outside the scope of this study. Nevertheless, the fact that RoPE states consistently outperform non-RoPE states *within the same model* across 5 different families strongly suggests the effect is driven by the encoding scheme itself.
>
> **Q4: Which features dominate top-k selection? Stability analysis?**
>
> **Response:** Our preliminary analysis indicates that **FFN activated states** often dominate the top-k selection simply due to their massive dimensionality (typically 4x model dimension). Furthermore, the high collinearity between transformer states makes feature-level attribution with Ridge Regression unstable. To rigorously address this, we are developing methods using group-lasso or similar sparsity-inducing penalties to handle collinearity, but we position this detailed feature-level interpretability as a crucial limitation to be addressed in future work.
>
> **Q5: Can targeted combination rules outperform generic concatenation?**
>
> **Response:** In the language network, winning states are distributed between **FFN Activated States** and **Per-Head Queries** (see Table 1 above). Developing a targeted rule is challenging because the "best state" map (Figure 1c in the manuscript) follows the complex, idiosyncratic folding of the cortex (gyri/sulci) rather than simple ROI boundaries. The generic concatenation + ridge regression approach (Mode 2) allows the model to learn these complex, voxel-specific weights data-drivenly, which likely outperforms rigid heuristic combination rules.

---

### Author Response · Authors · 2025-11-25
**General response to MindTransformer's reviewers**

We thank all the reviewers for their insightful and constructive feedback. To address all the concerns regarding per-subject analysis, generalization, baselines, and statistical controls, we have launched a comprehensive suite of new experiments. While the full-scale computation across all individual subjects and all 21 LLMs is ongoing, we will first report stable and robust preliminary results with the first 5 subjects out of 49 in the dataset based on Llama-3.2-1B, which is the model that performs the best in baselines, but modest in MindTransformer. We hope this could serve as a safe and temporary lower bound of the result compared to the whole 21 LLMs. We are now in the final phase of data analysis, paper revision, and response preparation. We will comment our response as soon as possible.

---

### Author Response · Authors · 2025-12-03
**General Response: Summary of Revisions**

We thank the Area Chair and all reviewers for the rigorous and constructive review process. The feedback has been crucial for refining our methodology and strengthening our claims.

Over the rebuttal period, we launched a comprehensive suite of new experiments to address the core concerns regarding **statistical rigor, generalization, and baseline comparisons**. We believe we have addressed the major limitations raised by all the reviewers.

Below is a summary of the key revisions and new evidence:

**1. Statistical Rigor and Baselines (Addressing Reviewers RBfX, c5Ks)**
* **Rigorous Baselines:** We introduced **Random Embedding** and **GloVe Static Embedding** baselines. MindTransformer consistently outperforms these controls (e.g., +46% improvement over GloVe in Heschl’s Gyrus), proving that our alignment captures specific structural and semantic signals, not just low-level timing or word onsets.
* **Dimensionality Control:** To address concerns about unfair feature comparisons (Reviewer RBfX), we conducted a controlled experiment equalizing feature dimensions across all states. Our results confirm that the functional dissociation (Attention with RoPE states for Auditory, FFN states for Language) holds even when dimensionality is strictly controlled, proving the gains are driven by information content, not parameter count.
* **Significance Testing:** We implemented a Voxel-wise Bootstrap Analysis with FDR correction ($q < 0.05$). As reported in our response to Reviewer c5Ks, our method maintains robust significance (>90% voxel coverage) across key auditory and language regions, confirming our results are not due to chance.

**2. Robustness and Generalization (Addressing Reviewers NyvA, c5Ks, 6Yhf)**
* **Per-Subject Analysis:** We moved beyond group averaging by reproducing our full analysis on individual subjects ($N=5$). The results confirm that our key findings—specifically the "attention layer $\\to$ Auditory Cortex" and "FFN $\\to$ Language Network" mappings—are robust and replicable at the individual level, despite the lower signal-to-noise ratio of single-subject fMRI.
* **Validation Protocol:** We clarified that **MindTransformer Mode 2** performs feature selection strictly within training folds, ensuring no data leakage from the test set.

**3. Theoretical Validation of Hierarchy (Addressing Reviewers 6Yhf, NyvA)**
* **Quantifying the Hierarchy:** We replaced qualitative observations with a quantitative **Computational Depth** metric. We demonstrated clear internal progression of the transformer block with the biological cortical depth along the auditory pathway. This provides strong evidence that the intra-block hierarchy mirrors the brain's sensory-to-association hierarchy.

**Conclusion**

We believe that **MindTransformer** offers a novel "Computational Neuroanatomy" framework that effectively bridges the gap between Transformer mechanics and neural representations. With the added statistical controls, per-subject validation, and calibrated claims, we are confident the manuscript now meets the high standards of ICLR.

---

### Meta-Review · Area_Chair_LCTa · 2026-01-05

**Summary:**

The reviewers' critical concerns can be summarized as follows:
- Reviewer NyvA - the strongest gains are confined to the low-level auditory cortex, while improvements in high-level language regions are modest.

- Reviewer aPhw - the methodology of linking brain activity and transformer representations primarily via correlation is debatable.

- Reviewer 6Yhf - several claims are perceived as mistated (e.g., claiming SOTA); the evaluation is limited to a single dataset; and some claims about hierarchy are insufficiently supported.

- Reviewer c5Ks - reproducibility and robustness at the individual-subject level are not adequately demonstrated.

- Reviewer RBfX - unfair comparisons across models with varying representational dimensionalities; the absence of stronger or more appropriate baselines weakens the empirical evaluation; and the analysis reports only winning ratios rather than raw performance values

**Reviewer Concerns:**

Most of the key concerns were successfully addressed in the rebuttal. However, three issues remain partially unresolved: (i) the improvements in high-level language regions are relatively modest, which raises questions about the overall conclusions, and underlying reasons for this pattern are not sufficiently explained beyond the strong performance of the baseline models; (ii) the reliance on a single dataset for evaluation, which, although it may not be feasible to address within this review cycle, remains a critical limitation; and (iii) the handling of representational dimensionality across different models requires a more principled treatment. More specifically, the AC would like to suggest the following action points to help strengthen the paper:

1. The modest improvements observed in high-level language regions - the authors should verify whether these gains are consistent when evaluated using alternative metrics and across additional datasets, in order to ensure the reliability of overall results

2. Dataset scope - while the authors provide justification for using a single fMRI dataset, the AC encourages them to follow the reviewers' recommendations by conducting additional experiments on other datasets that have already been suggested to the authors, rather than relying solely on the argument that a single dataset is sufficient.

3. Handling different dimensionality across the models - this concern would be better addressed by following the reviewers' suggestion to conduct experiments across models with varying intrinsic dimensionalities, rather than restricting dimensionality within a single model to analyze trends, as was done in the rebuttal.

Finally, the AC believes that the paper's overall contributions and strengths outweigh these remaining concerns, and that the work is therefore worthy of acceptance in this round. The AC encourages the authors to address the above remaining concerns more concretely in future revisions. As a minor note, the AC recommends that the authors treat MoE-based models and dense models separately in their analysis. Additionally, Figure 1 appears to incorrectly place the first LayerNorm inside the residual block rather than outside it.

**Reviewer Scores:**

Reviewer 6Yhf indicated they would be willing to raise their score from 6 to 8, and the AC believes the other reviewers may also be inclined to increase their scores as the discussion phase proceeds normally.

---

### Decision · Program_Chairs · 2026-01-26

Accept (Poster)